# SlotDiffusion: Object-Centric Generative Modeling with Diffusion Models

**Ziyi Wu**[1,2]  **Jingyu Hu**[1,*]  **Wuyue Lu**[1,*]  **Igor Gilitschenski**[1,2]  **Animesh Garg**[1,2]

[1]University of Toronto   [2]Vector Institute

## Abstract

Object-centric learning aims to represent visual data with a set of object entities (a.k.a. slots), providing structured representations that enable systematic generalization. Leveraging advanced architectures like Transformers, recent approaches have made significant progress in unsupervised object discovery. In addition, slot-based representations hold great potential for generative modeling, such as controllable image generation and object manipulation in image editing. However, current slot-based methods often produce blurry images and distorted objects, exhibiting poor generative modeling capabilities. In this paper, we focus on improving slot-to-image decoding, a crucial aspect for high-quality visual generation. We introduce SlotDiffusion – an object-centric Latent Diffusion Model (LDM) designed for both image and video data. Thanks to the powerful modeling capacity of LDMs, SlotDiffusion surpasses previous slot models in unsupervised object segmentation and visual generation across six datasets. Furthermore, our learned object features can be utilized by existing object-centric dynamics models, improving video prediction quality and downstream temporal reasoning tasks. Finally, we demonstrate the scalability of SlotDiffusion to unconstrained real-world datasets such as PASCAL VOC and COCO, when integrated with self-supervised pre-trained image encoders. Additional results and details are available at our website.

## 1 Introduction

Humans perceive the world by identifying discrete concepts such as objects and events [88], which serve as intermediate representations to support high-level reasoning and systematic generalization of intelligence [24]. In contrast, modern deep learning models typically represent visual data with patch-based features [14, 28], disregarding the compositional structure of scenes. Inspired by the human perception system, object-centric learning aims to discover the modular and causal structure of visual inputs. This approach thus holds the potential to improve the generalizability and interpretability of AI algorithms [50, 80]. For example, explicitly decomposing scenes into conceptual entities facilitates visual reasoning [8, 12, 13] and causal inference [80, 109]. Also, capturing the compositional structure of the world proves beneficial for both image generation [83, 90] and video prediction [100, 103].

However, unsupervised object-centric learning from raw perceptual input is challenging, as it requires capturing the appearance, position, and motion of each object. Earlier attempts at this problem usually bake in strong scene or object priors in their frameworks [19, 40, 55, 56], limiting their applicability to synthetic datasets [5, 42]. Later works extend the Scaled Dot-Product Attention mechanism [94] and introduce a general Slot Attention operation [60], which eliminates domain-specific priors. With a simple input reconstruction objective, these models learn to segment objects from raw images [60] and videos [46]. To date, state-of-the-art unsupervised slot models [83, 84] have shown promising scene decomposition results on naturalistic data under specialized scenarios such as traffic videos.

Despite tremendous progress in object segmentation, the generative capability of slot-based methods has been underexplored. For example, while the state-of-the-art object-centric model STEVE [84]

---

* Equal contribution

37th Conference on Neural Information Processing Systems (NeurIPS 2023).

is able to decompose complex data into slots, it reconstructs videos with severe object distortion and temporal inconsistency from slots [100]. Moreover, our own experiments reveal that all popular slot models [46, 60, 83] suffer from low slot-to-image decoding quality, preventing their adoption in visual generation such as image editing. Therefore, the goal of this work is to improve the generative capacity of object-centric models on complex data while preserving their segmentation performance.

We propose SlotDiffusion, an unsupervised object-centric model with a Latent Diffusion Model (LDM) [73] based slot decoder. We consider slots extracted from images as basic visual concepts (i.e., objects with different attributes), akin to text embeddings of phrases from sentences. The LDM decoder learns to denoise image feature maps guided by object slots, providing training signals for object discovery. Thanks to the expressive power of diffusion models, SlotDiffusion achieves a better trade-off between scene decomposition and visual generation. We evaluate our method on six synthetic and three real-world datasets consisting of complex images and videos. SlotDiffusion improves object segmentation results and achieves significant advancements in compositional generation. Crucially, we demonstrate that SlotDiffusion seamlessly integrates with recent development in object-centric learning, such as dynamics modeling [100] and the integration of pre-trained image encoders [81].

**Our main contributions** are as follows: **(i)** SlotDiffusion: a diffusion model based object-centric learning method. **(ii)** We apply SlotDiffusion to unsupervised object discovery and visual generation, where it achieves state-of-the-art results across both image and video datasets. **(iii)** We demonstrate that our learned slots can be directly utilized by the state-of-the-art object-centric dynamics model, improving future prediction and temporal reasoning performance. **(iv)** We further scale up SlotDiffusion to real-world datasets by integrating it with a self-supervised pre-trained image encoder.

## 2 Related Work

In this section, we provide a brief overview of related works on unsupervised object-centric learning and diffusion models, which is further expanded in Appendix B.

**Unsupervised object-centric learning** aims to discover the underlying structure of visual data and represent it with a set of feature vectors (a.k.a. slots). To achieve unsupervised scene decomposition, most methods use input reconstruction as the training signal. Earlier works often perform iterative inference to extract object slots, followed by a CNN-based decoder for reconstruction [6, 17, 23, 40, 55, 56]. AIR [19] and SQAIR [48] use a patch-based decoder to decode each object in a canonical pose, and then transform them back to the original position. Slot Attention [60] and SAVi [46] adopt the spatial broadcast decoder [97] to predict the RGB images and segmentation masks from each slot, which are combined via alpha masking. However, the per-slot CNN decoder limits the modeling capacity of these methods to synthetic datasets [5, 42]. Recently, SLATE [83] and STEVE [84] proposed a Transformer-based slot decoder. They first pre-train a dVAE to tokenize the input, and then train the slot-conditioned Transformer decoder to reconstruct patch tokens in an autoregressive manner. The powerful Transformer architecture and the feature-level reconstruction target enable them to segment naturalistic images and videos [79, 81, 85]. However, as observed in [83, 100], the slot-to-image reconstruction quality of Transformer decoders is low on complex data, sometimes even underperform CNN decoders. In this work, we leverage diffusion models [34, 73] as the slot decoder, which runs iterative denoising conditioned on slots. Thanks to their expressive power, our method not only improves scene decomposition, but also generates results with higher visual quality.

**Diffusion models** (DMs) [34, 87] have recently achieved tremendous progress in image generation [11, 64, 71, 77], showing their great capability in sample quality and input conditioning. The generative process of DMs is formulated as an iterative denoising procedure from Gaussian noise to clean data, where the denoiser is typically implemented as a U-Net [74]. However, the memory and computation requirements of DMs scale quadratically with the input resolution due to the self-attention layers in the U-Net. To reduce the training cost, LDM [73] proposes to first downsample images to feature maps with a pre-trained VAE encoder, and then run the diffusion process in this low-resolution latent space. LDM also introduces cross-attention as a flexible mechanism for conditional generation. For example, text-guided LDMs [73] perform cross-attention between text embeddings and U-Net's feature maps at multiple resolutions to guide the denoising process. In this work, we adopt LDM as the slot decoder due to its strong generation capacity, where the conditioning is achieved by cross-attention between the denoising feature maps and the object slots.

While conventional DMs excel in controllable generation such as image editing [31, 44] and compositional generation [57, 75], they often require specific supervision such as text to guide the generation

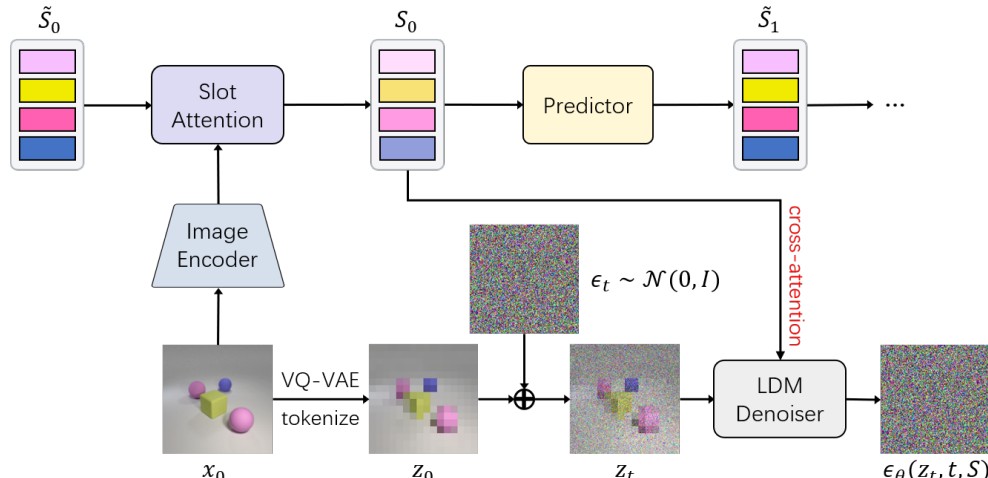

**Figure 1:** SlotDiffusion architecture overview. Given an initial frame of a video $x_0$, we initialize slots from a set of learnable vectors $\tilde{\mathcal{S}}_0$, and perform Slot Attention with image features to update object slots $\mathcal{S}_0$. During training, a U-Net denoiser predicts the noise $\epsilon_t$ added to the image tokens $z_0$ conditioned on slots via cross-attention. The entire model is applied recurrently to all video frames with a Predictor initializing future slots from current step.

process. In contrast, SlotDiffusion discovers meaningful visual concepts from data in an unsupervised manner, which can be used for controllable generation as will be shown in the experiments.

**Concurrent work.** Recently, Jiang et al. [41] also employ LDM in object-centric learning. However, they utilize an image tokenizer pre-trained on external datasets, while all components of our model are trained purely on target datasets. Moreover, their work only evaluates on images, while we test SlotDiffusion on both images and videos. In addition, we combine our model with recent development in object-centric learning, showing our potential in dynamics modeling and handling real-world data.

## 3   SlotDiffusion: Object-Oriented Diffusion Model

In this section, we describe SlotDiffusion by first reviewing previous unsupervised object-centric learning methods (Section 3.1). Then, we introduce our slot-conditioned diffusion model decoder (Section 3.2). Finally, we show how to perform compositional generation with the visual concepts learned by SlotDiffusion (Section 3.3). The overall model architecture is illustrated in Figure 1.

### 3.1   Background: Slot-based Object-Centric Learning

Unsupervised object-centric learning aims to represent a scene with a set of object *slots* without object-level supervision. Here, we review the SAVi family [46, 84] as they achieve state-of-the-art object discovery on videos. SAVi builds upon Slot Attention [60] and runs on videos in a recurrent encoder-decoder manner. Given $T$ input frames $\{\boldsymbol{x}_t\}_{t=1}^{T}$, SAVi first leverages an encoder to extract per-frame features, adds positional encodings, and flattens them into a set of vectors $\boldsymbol{h}_t = f_{\text{enc}}(\boldsymbol{x}_t) \in \mathbb{R}^{M \times D_{\text{enc}}}$. Then, the model initializes $N$ slots $\tilde{\mathcal{S}}_t \in \mathbb{R}^{N \times D_{\text{slot}}}$ from a set of learnable vectors ($t = 1$), and updates them with Slot Attention as $\mathcal{S}_t = f_{\text{SA}}(\tilde{\mathcal{S}}_t, \boldsymbol{h}_t)$. $f_{\text{SA}}$ performs soft feature clustering, where slots compete with each other to capture certain areas of the input via iterative attention [94]. To achieve temporally aligned slots, SAVi leverages a Transformer-based predictor to initialize $\tilde{\mathcal{S}}_t$ ($t \geq 2$) as $\tilde{\mathcal{S}}_t = f_{\text{pred}}(\mathcal{S}_{t-1})$. Finally, the model utilizes a decoder $f_{\text{dec}}$ to reconstruct input $\boldsymbol{x}_t$ from slots $\mathcal{S}_t$ and uses the reconstruction loss as training signal. Below we detail the slot decoder $f_{dec}$ studied in previous works as we focus on improving it in this paper, and refer the readers to [46] for more information about other model components. See Appendix A for figures of different slot decoders.

**Mixture-based decoder.** Vanilla SAVi [46] adopts the Spatial Broadcast decoder [97] consisting of a stack of up-sampling deconvolution layers. It first broadcasts each slot $\mathcal{S}_t^i$ to a 2D grid to form a feature map. Each feature map is then decoded to an RGB image $\boldsymbol{y}_t^i$ and an alpha mask $\boldsymbol{m}_t^i$, which are combined into the final reconstructed image $\hat{\boldsymbol{x}}_t$. SAVi is trained with an MSE reconstruction loss:

$$(\boldsymbol{y}_t^i, \boldsymbol{m}_t^i) = f_{\text{dec}}^{\text{mix}}(\mathcal{S}_t^i), \quad \hat{\boldsymbol{x}}_t = \sum_{i=1}^{N} \boldsymbol{m}_t^i \odot \boldsymbol{y}_t^i, \quad \mathcal{L}_{\text{image}} = \sum_{t=1}^{T} ||\boldsymbol{x}_t - \hat{\boldsymbol{x}}_t||^2. \tag{1}$$

**Transformer-based decoder.** The above mixture-based decoder has limited modeling capacity as it decodes each slot separately without interactions. Also, pixel-level reconstruction biases the model to low-level color statistics, which only proves effective on objects with uniform colors, and cannot

scale to complex data with textured objects. Current state-of-the-art model STEVE [84] thus proposes to reconstruct intermediate features produced by another network [83]. Given frame $\boldsymbol{x}_t$, STEVE leverages a pre-trained dVAE encoder to convert it into a sequence of patch tokens $\boldsymbol{o}_t = \{\boldsymbol{o}_t^i\}_{i=1}^L$, which serve as the reconstruction targets for the autoregressive Transformer decoder:

$$\boldsymbol{o}_t = f_{\text{enc}}^{\text{dVAE}}(\boldsymbol{x}_t), \;\; \hat{\boldsymbol{o}}_t^l = f_{\text{dec}}^{\text{trans}}(\mathcal{S}_t; \boldsymbol{o}_t^1, ..., \boldsymbol{o}_t^{l-1}), \;\; \mathcal{L}_{\text{token}} = \sum_{t=1}^T \sum_{l=1}^L \text{CrossEntropy}(\boldsymbol{o}_t^l, \hat{\boldsymbol{o}}_t^l). \quad (2)$$

Thanks to the cross-attention mechanism in the Transformer decoder and the feature-level reconstruction objective, STEVE works on naturalistic videos with textured objects and backgrounds. However, the generation quality of the Transformer slot decoder is still low on complex data [83, 100].

## 3.2 Slot-conditioned Diffusion Model

Object-centric generative models [90, 110] often decompose the generation process to first predicting object slots in the latent space, and then decoding slots back to the pixel space. Here, the image-to-slot abstraction and the slot-to-image decoding are often performed by a pre-trained object-centric model. Therefore, the generation quality is greatly affected by the scene decomposition result and the slot decoder capacity. As observed in [83, 100], methods with mixture-based decoders usually generate images with blurry objects, as they fail to capture objects from complex data. While models with Transformer-based decoders are better at scene decomposition, they still produce results of low visual quality. We attribute this to two issues: **(i)** treating images as sequences of tokens ignores their inherent spatial structure; **(ii)** autoregressive token prediction causes severe error accumulation. In this work, we overcome these drawbacks by introducing diffusion models as the slot decoder, which preserves the spatial dimension of images, and iteratively refines the generation results. From now on, we omit $t$ as the video timestamp, and will only use $t$ to denote the timestep in the diffusion process.

**Diffusion model.** Diffusion models (DMs) [34, 87] are probabilistic models that learn a data distribution $p_\theta(\boldsymbol{X}_0)$ by gradually denoising a standard Gaussian distribution, in the form of $p_\theta(\boldsymbol{X}_0) = \int p_\theta(\boldsymbol{X}_{0:T}) \, d\boldsymbol{X}_{1:T}$, where $\boldsymbol{X}_{1:T}$ are intermediate denoising results. The forward process of DMs is a Markov Chain that iteratively adds Gaussian noise to the clean data $\boldsymbol{X}_0$, which is controlled by a pre-defined variance schedule $\{\beta_t\}_{t=1}^T$. Let $\alpha_t = 1 - \beta_t$ and $\bar{\alpha}_t = \prod_{s=1}^t \alpha_s$, we have:

$$q(\boldsymbol{X}_t|\boldsymbol{X}_{t-1}) = \mathcal{N}(\boldsymbol{X}_t|\sqrt{1-\beta_t}\boldsymbol{X}_{t-1}, \beta_t\boldsymbol{I}) \;\Rightarrow\; q(\boldsymbol{X}_t|\boldsymbol{X}_0) = \mathcal{N}(\boldsymbol{X}_t|\sqrt{\bar{\alpha}_t}\boldsymbol{X}_0, (1-\bar{\alpha}_t)\boldsymbol{I}). \quad (3)$$

During training, a denoising model $\epsilon_\theta(\boldsymbol{X}_t, t)$ is trained to predict the noise applied to a noisy sample:

$$\boldsymbol{X}_t = \sqrt{\bar{\alpha}_t}\boldsymbol{X}_0 + \sqrt{1-\bar{\alpha}_t}\epsilon_t, \;\; \mathcal{L}_{\text{DM}} = ||\epsilon_t - \epsilon_\theta(\boldsymbol{X}_t, t)||^2, \text{ where } \epsilon_t \sim \mathcal{N}(\boldsymbol{0}, \boldsymbol{I}). \quad (4)$$

At inference time, we can start from a random Gaussian noise, and apply the trained denoiser to iteratively refine the sample. See Appendix C for a detailed formulation of diffusion models.

**SlotDiffusion.** Our model consists of the same model components as SAVi and STEVE, while only replacing the slot decoder with the DM-based one. Inspired by the success of feature-level reconstruction in STEVE, we adopt the Latent Diffusion Model (LDM) [73] as the decoder, which denoises features in the latent space. This improves the segmentation results with a higher-level reconstruction target, and greatly reduces the training cost as it runs at a lower resolution. Specifically, we pre-train a VQ-VAE [72] to extract feature maps $\boldsymbol{z} \in \mathbb{R}^{h \times w \times D_{\text{vq}}}$ from $\boldsymbol{x}$ before training SlotDiffusion:

$$\boldsymbol{z} = f_{\text{enc}}^{\text{VQ}}(\boldsymbol{x}), \;\; \hat{\boldsymbol{x}} = f_{\text{dec}}^{\text{VQ}}(\boldsymbol{z}), \;\; \mathcal{L}_{\text{VQ}} = ||\boldsymbol{x} - \hat{\boldsymbol{x}}||^2 + \text{LPIPS}(\boldsymbol{x}, \hat{\boldsymbol{x}}), \quad (5)$$

where $\text{LPIPS}(\cdot, \cdot)$ is the VGG perceptual loss [108]. In preliminary experiments, we also tried KL-VAE [73] and VQ-GAN [20] as the image tokenizer, but did not observe clear improvements.

To condition the LDM decoder on slots $\mathcal{S}$, we notice that slots are a set of $N$ feature vectors, which are similar to text embeddings output by language models [69, 70]. Therefore, we follow recent text-guided LDMs [73] to guide the denoising process via cross-attention:

$$\boldsymbol{c} = \text{CrossAttention}(Q(\tilde{\boldsymbol{c}}), K(\mathcal{S}), V(\mathcal{S})). \quad (6)$$

Here, $Q$, $K$, $V$ are learnable linear projections, $\tilde{\boldsymbol{c}}$ is an intermediate feature map from the denoising U-Net $\epsilon_\theta$, and $\boldsymbol{c}$ is the feature map updated with slot information. An important property of cross-attention is that the result in Eq. (6) is order-invariant to the conditional input $\mathcal{S}$, thus preserving the permutation-equivariance of slots, which is a key property of object-centric models. In practice, we perform conditioning after several layers in the U-Net at multiple resolutions. Overall, our model is trained with a slot-conditioned denoising loss over VQ-VAE feature maps:

$$\boldsymbol{z}_t = \sqrt{\bar{\alpha}_t}\boldsymbol{z} + (1-\bar{\alpha}_t)\epsilon_t, \;\; \mathcal{L}_{\text{slot}} = ||\epsilon_t - \epsilon_\theta(\boldsymbol{z}_t, t, \mathcal{S})||^2, \text{ where } \epsilon_t \sim \mathcal{N}(\boldsymbol{0}, \boldsymbol{I}). \quad (7)$$

### 3.3 Compositional Generation with Visual Concepts

Previous conditional DMs [71, 73, 77] rely on labels such as text to control the generation process. In contrast, SlotDiffusion is conditioned on slots output by Slot Attention, which is trained jointly with the DM. We consider each object slot extracted from an image as an instance of a basic visual concept (e.g., red cube). Therefore, we can discover visual concepts from unlabeled data, and build a library of slots for each of them. Then, we can compose concepts via slots to generate novel samples.

**Visual concept library.** We follow SLATE [83] to build visual concept libraries by clustering slots. Take image datasets for example, we first collect slots extracted from all training images, and then apply K-Means clustering to discover $K$ clusters, each serves as a concept. All slots assigned to a cluster form a library for that concept. On video datasets, we discover concepts by clustering slots extracted from the first frame. As shown in our experiments, SlotDiffusion discovers semantically meaningful concepts, such as objects with different shapes, and different components of human faces.

To generate new samples, we first select $N$ concepts from $K$ candidates, pick one slot from each concept's library, and then decode them with our LDM-based slot decoder. To avoid severe occlusions, we reject slots with object segmentation masks overlapping greater than an mIoU threshold.

## 4 Experiments

SlotDiffusion is a generic unsupervised object-centric learning framework that is applicable to both image and video data. We conduct extensive experiments to answer the following questions: **(i)** Can we learn object-oriented scene decomposition supervised by the denoising loss of DMs? (Section 4.2) **(ii)** Will our LDM-based decoder improve the visual generation quality of slot models? (Section 4.3) **(iii)** Is the object-centric representation learned by SlotDiffusion useful for downstream dynamics modeling tasks? (Section 4.4) **(iv)** Can we extend our method to handle real-world data? (Section 4.5) **(v)** Can SlotDiffusion benefit from other recent improvements in object-centric learning? (Section 4.6) **(vi)** What is the impact of each design choice on SlotDiffusion? (Section 4.7)

### 4.1 Experimental Setup

**Datasets.** We evaluate our method in unsupervised object discovery and slot-based visual generation on six datasets, namely, the two most complex image datasets *CLEVRTex* [43] and *CelebA* [59] from SLATE [83], and four video datasets *MOVi-D/E/Solid/Tex* [25] from STEVE [84]. Then, we show SlotDiffusion's capability for downstream video prediction and reasoning tasks on *Physion* [4]. Finally, we scale our method to unconstrained real-world images on *PASCAL VOC 2012* [21] and *MS COCO 2017* [54]. We briefly introduce each dataset below and provide more details in Appendix D.1.

*CLEVRTex* augments the CLEVR [42] dataset with more diverse object shapes, materials, and textures. The backgrounds also present complex textures compared to the plain gray one in CLEVR. We train our model on the training set with 40k images, and test on the test set with 5k samples.

*CelebA* contains over 200k real-world celebrity images. All images are mainly occupied by human faces, with roughly front-view head poses. This dataset is more challenging as real-world images typically have background clutter and complicated lighting conditions. We train our model on the training set with around 160k images, and test on the test split with around 20k images.

*MOVi-D/E/Solid/Tex* consist of videos generated using the Kubric simulator [25]. Their videos feature photo-realistic backgrounds and diverse real-world objects, where one or several objects are thrown to the ground to collide with other objects. We follow the official train-test split to evaluate our model.

*Physion* is a VQA dataset containing realistic simulations of eight physical scenarios, such as dropping and soft-body deformation. The goal of Physion is to predict whether two target objects will contact as the scene evolves. Following the official protocol, we first train models to learn scene dynamics and evaluate the video prediction results, and then perform VQA on the model's future rollout.

*PASCAL VOC 2012/MS COCO 2017* are real-world datasets commonly used in object detection and segmentation. They are more challenging than CelebA as the images capture unconstrained natural scenes with multiple objects of varying sizes. We will denote them as *VOC* and *COCO* for short.

**Baselines.** We compare SlotDiffusion with state-of-the-art fully unsupervised object-centric models. On image datasets, we adopt *Slot Attention* (SA for short) [60] and *SLATE* [83]. On video datasets, we adopt *SAVi* [46] and *STEVE* [84]. They are representative models which use the mixture-based

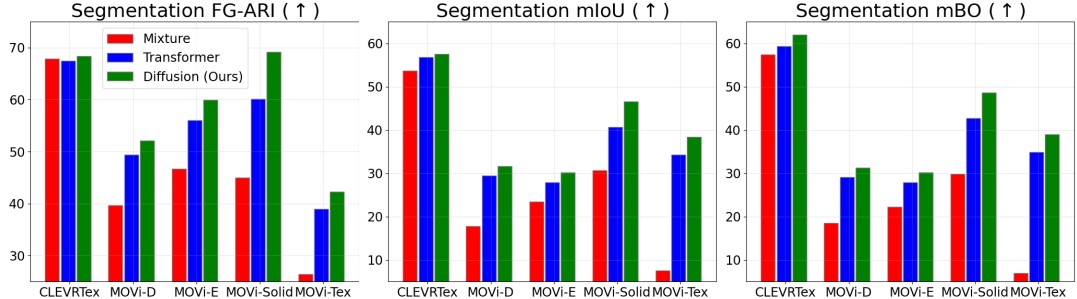

**Figure 2:** Unsupervised object segmentation measured by FG-ARI (left), mIoU (middle), and mBO (right). Mixture, Transformer, and Diffusion stand for SA/SAVi, SLATE/STEVE, and SlotDiffusion, respectively.

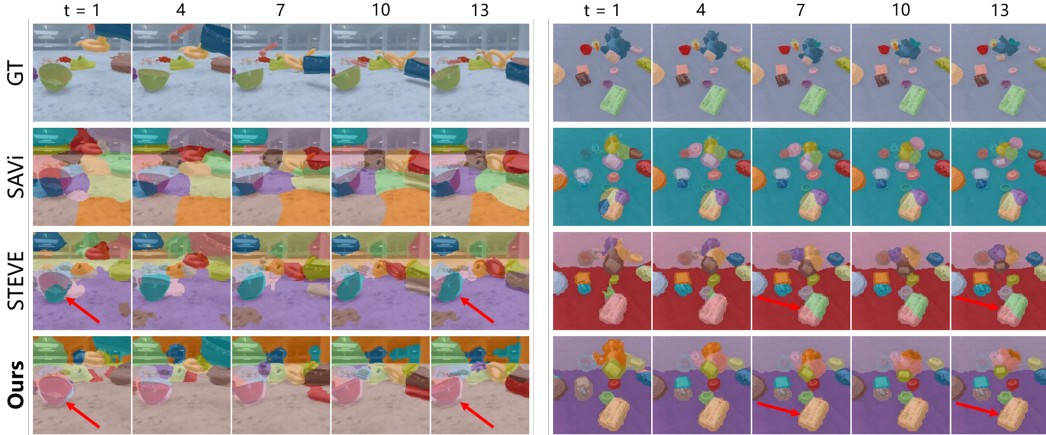

**Figure 3:** Visualization of video segmentation results on MOVi-D (left) and MOVi-E (right).

decoder and the Transformer-based decoder. We will introduce other baselines in each task below. See Appendix D.3 for implementation details of all baselines adopted in this paper.

**Our Implementation Details.** We use the same image encoder, Slot Attention module, and transition predictor as baselines, while only replacing the slot decoder with the conditional LDM. We first pre-train VQ-VAE on each dataset, and then freeze it and train the object-centric model with Eq. (7). Please refer to Appendix D.4 for complete training details, and Appendix E.1 for a comparison of computation requirements of SlotDiffusion and baseline methods.

## 4.2 Evaluation on Object Segmentation

**Evaluation Metrics.** Following previous works, we use foreground Adjusted Rand Index (*FG-ARI*) to measure how well the predicted object masks match the ground-truth segmentation. As pointed out by [17, 43], FG-ARI is sub-optimal since it only considers foreground pixels and ignores false segmentation of backgrounds. Therefore, we also adopt the mean Intersection over Union (*mIoU*) and mean Best Overlap (*mBO*) computed over all pixels. See Appendix D.2 for metric implementations.

**Results.** Figure 2 presents the results on unsupervised object segmentation. We do not evaluate on CelebA as it does not provide object mask annotations. SlotDiffusion outperforms baselines in three metrics across all datasets, proving that the denoising loss in LDMs is a good training signal for scene decomposition. Compared to CLEVRTex, our method achieves larger improvements on video datasets since SlotDiffusion also improves the tracking consistency of objects. Figure 3 shows a qualitative result of video segmentation. As observed in previous work [84], MOVi-E's moving camera provides motion cues compared to static cameras in MOVi-D, making object discovery easier. Indeed, SAVi degenerates to stripe patterns on MOVi-D, but is able to produce meaningful masks on MOVi-E. Compared to STEVE, our method has fewer object-sharing issues (one object being captured by multiple slots), especially on large objects. See Appendix E.2 for more qualitative results.

## 4.3 Evaluation on Generation Capacity

**Evaluation Metrics.** To generate high-quality results, object-centric generative models need to first represent objects with slots faithfully, and then compose them in novel ways. Hence, we first evaluate slot-to-image reconstruction, which inspects the expressive power of object slots. We adopt *MSE* and *LPIPS* [108] following previous works. As discussed in [100, 108], LPIPS better aligns with human perception, while MSE favors blurry images. We thus focus on comparing LPIPS, and report MSE

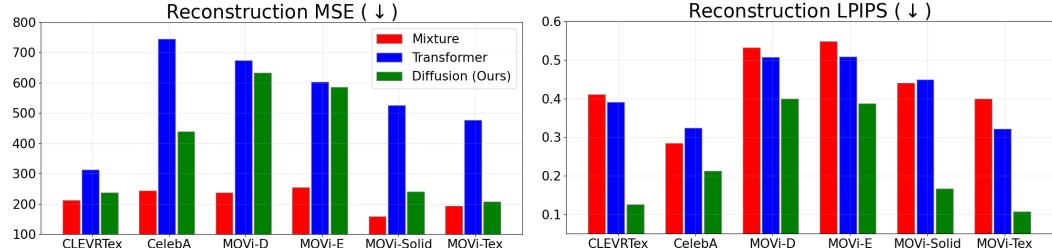

**Figure 4:** Slot-based image and video reconstruction measured by MSE (left) and LPIPS (right).

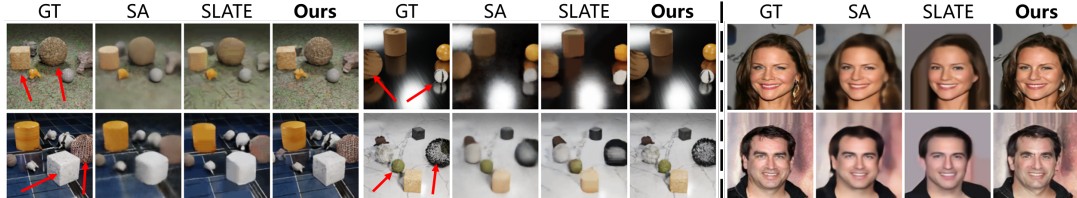

**Figure 5:** Reconstruction results on CLEVRTex (left) and CelebA (right). Our method better preserves the textures on objects and backgrounds in CLEVRTex, and the human hairs and facial features in CelebA.

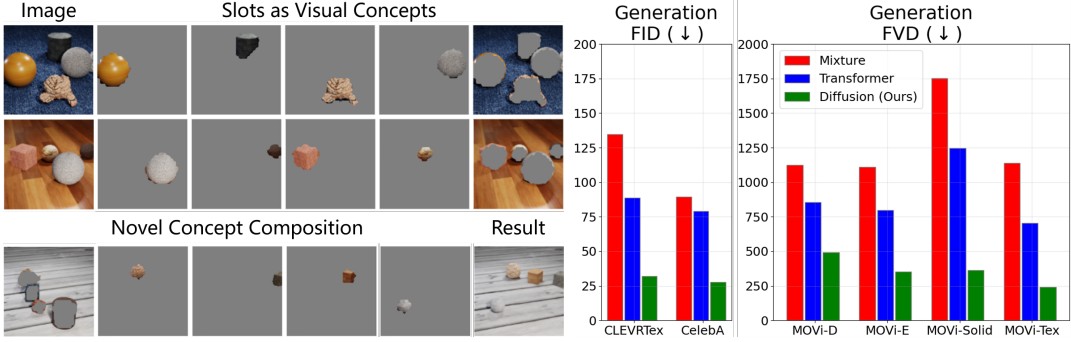

(a) Visual concepts discovery and composition      (b) Quantitative results on compositional generation

**Figure 6:** The mechanism of slot-based compositional generation (a) and the quantitative results (b). Our method first discovers visual concepts from images in the form of slots, such as objects and background maps in CLEVRTex. Then, we can synthesize novel samples by composing concept slots from different images.

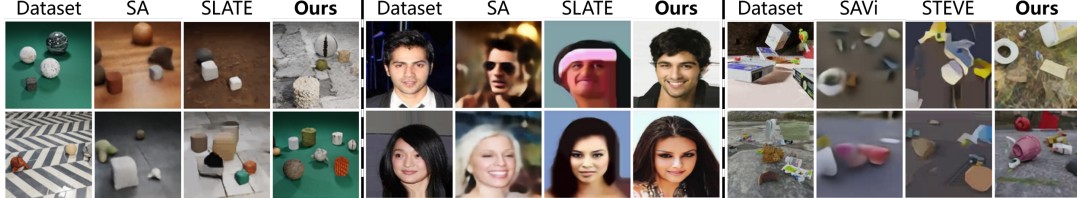

**Figure 7:** Compositional generation samples on CLEVRTex (left), CelebA (middle), and MOVi-E (right). We randomly select slots from the visual concept libraries built from training data, and decode them into images or videos. SlotDiffusion synthesizes higher-fidelity results with more details compared to baselines.

for completeness. In addition to reconstruction, we conduct compositional generation with the visual concepts learned in Section 3.3. *FID* [32] and *FVD* [92] are used to measure the generation quality.

**Results on reconstruction.** Figure 4 presents the reconstruction results. SlotDiffusion outperforms both baselines with a sizeable margin in LPIPS on all datasets, and achieves the second best MSE. As shown in the qualitative results in Figure 5, SA with a mixture-based decoder produces blurry images despite a lower MSE. Also, the Transformer-based decoder reconstructs distorted objects, and loses all the textures on object surfaces and backgrounds. In contrast, our LDM-based decoder is able to iteratively refine the results, leading to accurate textures and details such as human hairs.

**Results on compositional generation.** As shown in Figure 6 (a), we discover visual concepts by clustering slots extracted from images, and randomly composing slots from each concept's library to generate novel samples. Figure 6 (b) summarizes the quantitative results. SlotDiffusion significantly improves the FID and FVD scores, since it synthesizes images and videos with coherent contents and much better details compared to baselines. This is verified by the qualitative results in Figure 7. See Appendix E.3 for more visualizations of compositional generation results.

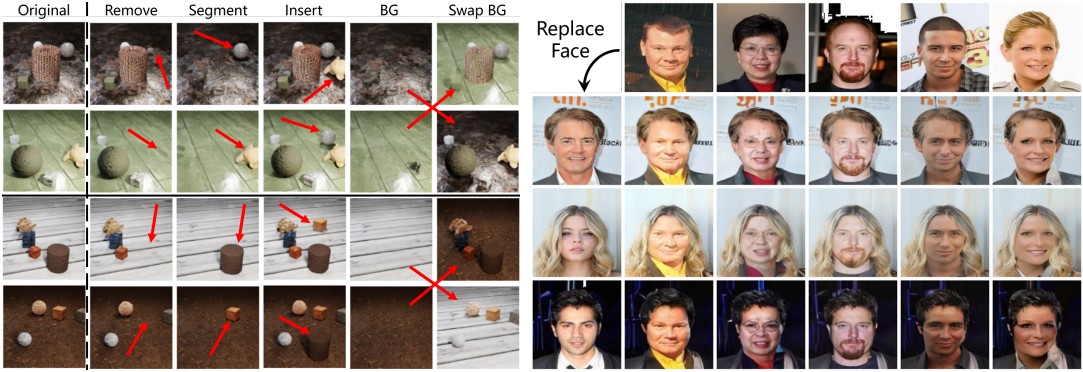

| (a) Scene editing on CLEVRTex | (b) Face replacement on CelebA |

**Figure 8:** Slot-based image editing. *Left*: we can manipulate objects, retrieve and swap background maps in a scene. *Right*: we can replace the human faces in the first column with the faces from the first row.

**Table 1:** Video prediction results on Physion.

| Method | MSE | LPIPS | FVD |
|---|---|---|---|
| PredRNN | **286.3** | 0.38 | 756.7 |
| VQFormer | 619.0 | 0.33 | 694.5 |
| STEVE + SlotFormer | 832.0 | 0.43 | 930.6 |
| **Ours** + SlotFormer | 477.7 | **0.26** | **582.2** |

**Table 2:** Physion VQA accuracy. We report observation (Obs.), observation plus rollout (Dyn.), and the improvement (↑).

| Method | Obs. (%) | Dyn. (%) | ↑ (%) |
|---|---|---|---|
| RPIN | 62.8 | 63.8 | +1.0 |
| pDEIT-lstm | 59.2 | 60.0 | +0.8 |
| STEVE + SlotFormer | 65.2 | 67.1 | +1.9 |
| **Ours** + SlotFormer | **67.5** | **69.7** | **+2.2** |

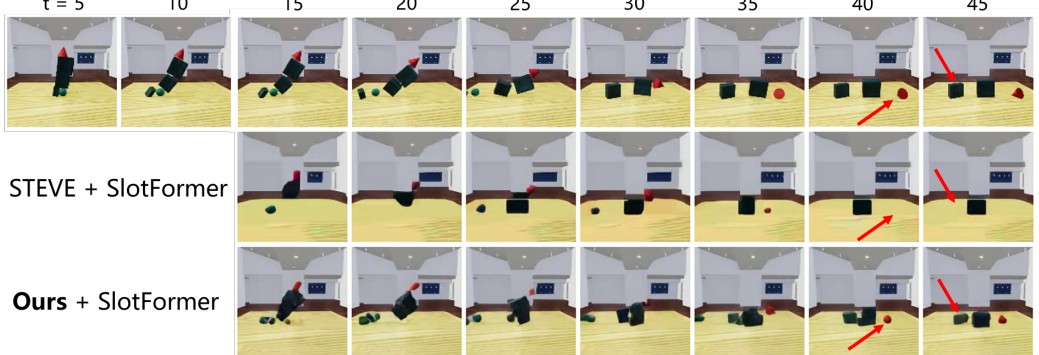

**Figure 9:** Qualitative results of video prediction on Physion. The first row shows ground-truth frames, while the other rows show the model rollout results. See Appendix E.4 for visualizations of all baselines.

**Slot-based image editing.** In Figure 8, we apply SlotDiffusion to image editing by manipulating slots. On CLEVRTex, we can remove an object by segmenting it out, and insert it to another scene. We can also extract background maps from an image and swap it with another image. On CelebA, SlotDiffusion decomposes images to faces, hairs, clothes, and backgrounds. By swapping the face slot, our method generates realistic images with new faces, while keeping other components unchanged.

### 4.4 Downstream Tasks: Video Prediction and VQA

In this subsection, we evaluate the object slots learned by SlotDiffusion in downstream tasks. We select Physion dataset as it requires capturing the motion and physical properties of objects for accurate dynamics simulation. We train the state-of-the-art object-centric dynamics model SlotFormer [100] over slots extracted by our model, and unroll it to perform video prediction and question answering.

**Video prediction baselines.** We adopt three baselines that build upon different image feature representations: *PredRNN* [96] that uses global image feature maps, *VQFormer* which utilizes image patch tokens encoded by VQ-VAE [72], and *STEVE + SlotFormer* that runs the same SlotFormer dynamics model over object-centric slot representations extracted by STEVE.

**Results on video prediction.** Table 1 presents the results on the visual quality of the generated videos. SlotDiffusion with SlotFormer outperforms baselines clearly in LPIPS and FVD, and achieves a competitive result on MSE. As discussed in Section 4.3 and [100], MSE is a poor metric in generation. Figure 9 visualizes the model rollouts. Compared to STEVE with SlotFormer, our method generates videos with higher fidelity such as the consistent boxes and the red object.

**Table 3:** Unsupervised object segmentation results on real-world datasets: VOC (left) and COCO (right).

| PASCAL VOC | FG-ARI | mBO$^i$ | mBO$^c$ | MS COCO | FG-ARI | mBO$^i$ | mBO$^c$ |
|---|---|---|---|---|---|---|---|
| SA + DINO ViT | 12.3 | 24.6 | 24.9 | SA + DINO ViT | 21.4 | 17.2 | 19.2 |
| SLATE + DINO ViT | 15.6 | 35.9 | 41.5 | SLATE + DINO ViT | 32.5 | 29.1 | 33.6 |
| DINOSAUR | **23.2** | 43.6 | 50.8 | DINOSAUR | 34.3 | **32.3** | **38.8** |
| **Ours** + DINO ViT | 17.8 | **50.4** | **55.3** | **Ours** + DINO ViT | **37.2** | 31.0 | 35.0 |

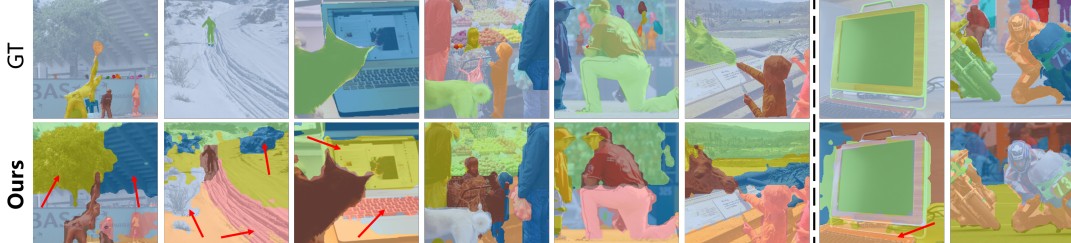

**Figure 10:** Segmentation results on real-world images: COCO (left) and VOC (right). SlotDiffusion is able to discover unannotated but meaningful regions highlighted by the red arrows, such as trees and keyboards.

**Table 4:** Unsupervised object segmentation results of SlotDiffusion with other improvements on CLEVRTex.

| Method | FG-ARI | mIoU | mBO |
|---|---|---|---|
| SlotDiffusion | 68.39 | 57.56 | 61.94 |
| SlotDiffusion + Slot-TTA | **69.66** | **58.67** | **62.82** |

| Method | FG-ARI | mIoU | mBO |
|---|---|---|---|
| BO-QSA | **80.47** | 64.26 | 66.27 |
| SlotDiffusion | 68.39 | 57.56 | 61.94 |
| BO-SlotDiffusion | 78.50 | 64.35 | **68.68** |

**VQA baselines.** We select two baselines following [100]. *RPIN* is an object-centric dynamics model with ground-truth object bounding box information. *pDEIT-lstm* builds LSTM over frame features extracted by an ImageNet [10] pre-trained DeiT [91]. We also compare with *STEVE + SlotFormer*.
**Results on VQA.** Table 2 summarizes the VQA accuracy on observation only (Obs.) and observation plus rollout frames (Dyn.). SlotDiffusion with SlotFormer achieves the best result in both cases, with the highest rollout gain. This proves that our method learns informative slots which capture correct features for learning object dynamics, such as the motion of objects falling down in Figure 9.

### 4.5 Scaling to Real-World Data

To show the scalability of our method to real-world data, we follow [81] to leverage pre-trained models. We replace the image encoder in SlotDiffusion with a self-supervised DINO [7] pre-trained ViT encoder. The model is still trained with the LDM denoising loss over VQ-VAE image tokens.

**Baselines.** To directly compare the effect of the slot decoders, we adopt SA and SLATE with the same DINO pre-trained ViT image encoder. We also compare with DINOSAUR [81] with the same image encoder, which reconstructs the ViT feature maps instead of input images as training signal.

**Results.** Table 3 presents the segmentation results on VOC and COCO. We follow [81] to report mBO$^i$ and mBO$^c$, which are mBOs between predicted masks and ground-truth instance and semantic segmentation masks. SlotDiffusion consistently outperforms SA and SLATE over all metrics, proving the effectiveness of our LDM-based slot decoder. Our model is also competitive with the state-of-the-art method DINOSAUR. The performance variations may be because COCO has more instances per image than VOC. As shown in the qualitative results in Figure 10, SlotDiffusion is able to segment the major objects in an image, and discover unannotated but semantically meaningful regions.

### 4.6 Combining SlotDiffusion with Recent Improvements in Slot-based Models

As an active research topic, there have been several recent improvements on slot-based models [39, 67, 85]. Since these ideas are orthogonal to our modifications to the slot decoder, we can incorporate them into our framework to further boost the performance.

**Slot-TTA** [67] proposes a test-time adaptation method by optimizing the reconstruction loss of slot models on testing images. Similarly, we optimize the denoising loss of SlotDiffusion at test time. As shown in Table 4 left, Slot-TTA consistently improves the segmentation results of our method.

**BO-QSA** [39] applies bi-level optimization in the iterative slot update process. We directly combine it with SlotDiffusion (dubbed *BO-SlotDiffusion*). As shown in Table 4 right, BO-SlotDiffusion greatly

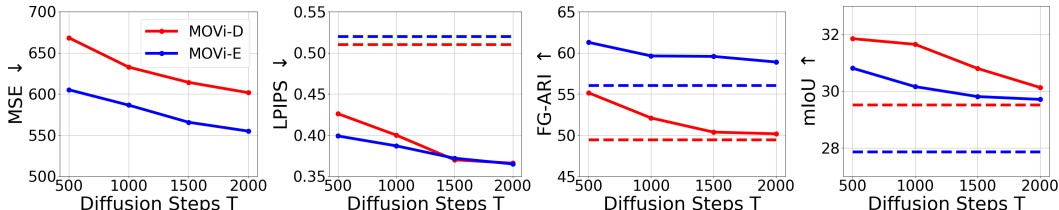

**Figure 11:** Ablation study on the number of diffusion steps $T$ of SlotDiffusion. We show the reconstruction performance (MSE, LPIPS) and segmentation results (FG-ARI, mIoU) on MOVi-D and MOVi-E. We also plot the *best baseline results* on each dataset as dashed lines. We do not plot MSE as it favors blurry generations.

improves the segmentation results compared to vanilla SlotDiffusion, and achieves comparable performance with BO-QSA. Note that the generation quality of BO-QSA is close to vanilla Slot Attention, and thus both our methods are significantly better than it.

### 4.7 Ablation Study

We study the effect of the number of diffusion process steps $T$ in our LDM-based decoder. We plot both the reconstruction and segmentation results on MOVi-D and MOVi-E in Figure 11. As expected, using more denoising steps leads to better generation quality. Interestingly, smaller $T$ results in better segmentation performance. This indicates that there is a trade-off between the reconstruction and segmentation quality of our model, and we select $T = 1000$ to strike a good balance. Literature on self-supervised representation learning [29, 38] suggests that a (reasonably) more difficult pretext task usually leads to better representations. Analogously, we hypothesize that a smaller $T$ makes the denoising pretext task harder, resulting in better object-centric representations.

Overall, SlotDiffusion outperforms baselines consistently across different $T$ in LPIPS, mIoU, and FG-ARI, proving that our method is robust against this hyper-parameter.

## 5  Conclusion

In this paper, we propose SlotDiffusion by introducing diffusion models to object-centric learning. Conditioned on object slots, our LDM-based decoder performs iterative denoising over latent image features, providing strong learning signals for unsupervised scene decomposition. Experimental results on both image and video datasets demonstrate our advanced segmentation and visual generation performance. Moreover, SlotDiffusion can be combined with recent progress in object-centric models to achieve state-of-the-art video prediction and VQA results, and scale up to real-world data. We briefly discuss the limitations and future directions here, which are further extended in Appendix F.

**Limitations.** With the help of pre-trained encoders such as DINO ViT [7], SlotDiffusion is able to segment objects from naturalistic images [21, 54]. However, we are still unable to decode natural images faithfully from slots. In addition, objects in real-world data are not well-defined. For example, the part-whole hierarchy causes ambiguity in the segmentation of articulated objects, and the best way of decomposition depends on the downstream task. Finally, our learned slot representations are still entangled, while many tasks require disentanglement between object positions and semantics.

**Future Works.** From the view of generative modeling, we can explore better DM designs to improve the modeling capacity of local details. To generate high-quality natural images, one solution is to incorporate pre-trained generative models such as the text-guided Stable Diffusion [73]. From the view of object-centric learning, we want to apply SlotDiffusion to more downstream tasks such as reinforcement learning [95, 105] and 3D modeling [86, 104]. It is interesting to see if task-specific supervision signals (e.g., rewards in RL) can lead to better scene decomposition.

## Broader Impacts

Unsupervised object-centric learning is an important task in machine learning and computer vision. Equipping machines with a structured latent space enables more interpretable and robust AI algorithms, and can benefit downstream tasks such as scene understanding and robotics. We believe this work benefits both the research community and the society.

**Potential negative societal impacts.** We do not see significant risks of human rights violations or security threats in our work. However, since our method can be applied to generation tasks, it might generate unpleasing content. Future research should also avoid malicious uses such as social surveillance, and be careful about the training cost of large diffusion models.

## Acknowledgments

We would like to thank Gautam Singh, Maximilian Seitzer for the help with data preparation and discussions about technical details of baselines, Xuanchi Ren, Tianyu Hua, Wei Yu, all colleagues from TISL for fruitful discussions about diffusion models, and Xuanchi Ren, Liquan Wang, Quincy Yu, Calvin Yu, Ritviks Singh for support in computing resources. We would also like to thank the anonymous reviewers for their constructive feedback. Animesh Garg is supported by CIFAR AI chair, NSERC Discovery Award, University of Toronto XSeed Grant, and NSERC Exploration grant. We would like to acknowledge Vector Institute for computation support.

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

# A  Model Architecture

Figure 12 shows the general pipeline of video Slot Attention models, and compare two existing slot decoders with our proposed Latent Diffusion Model (LDM) [73] based decoder.

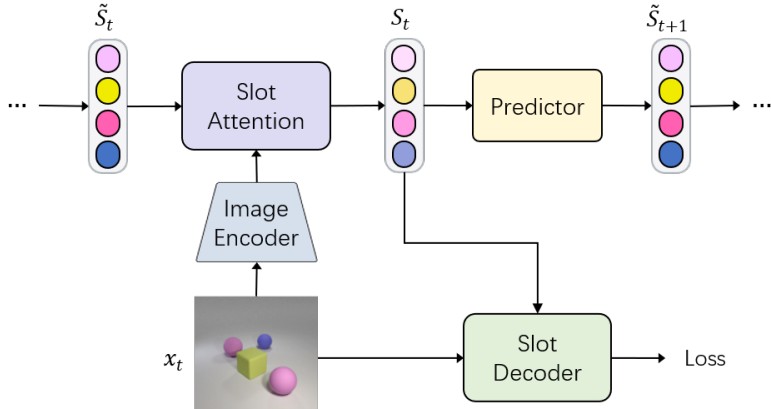

(a) General pipeline of video-based Slot Attention models.

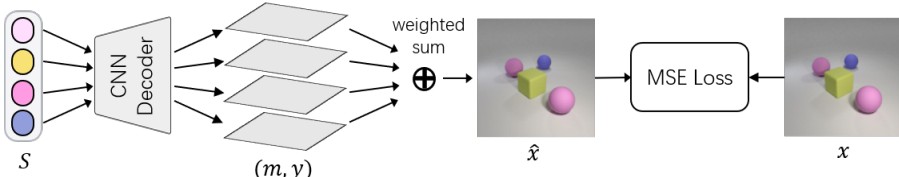

(b) Mixture-based CNN decoder.

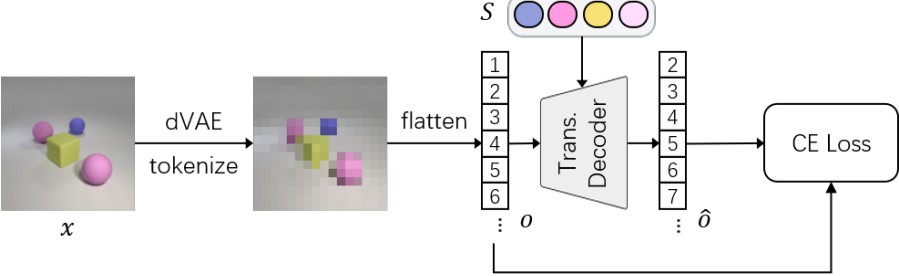

(c) Transformer-based autoregressive decoder.

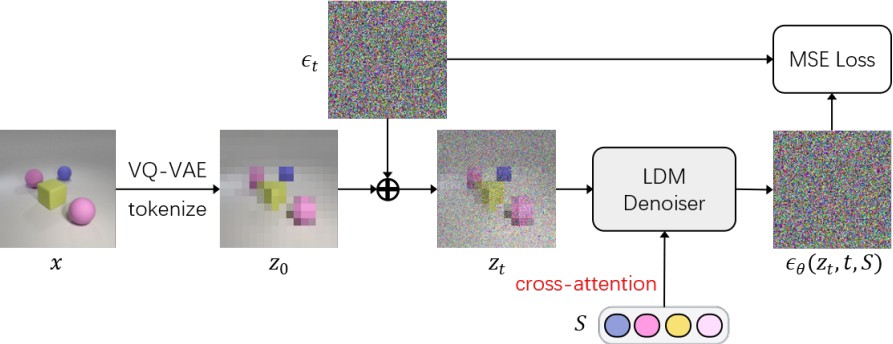

(d) Latent diffusion model based decoder.

**Figure 12:** Illustration of (a) the training pipeline of video Slot Attention models, (b) SAVi's mixture-based decoder [46], (c) STEVE's Transformer-based decoder [84], and (d) our LDM-based decoder.

## B  Additional Related Work

**Unsupervised object-centric learning from images.** Our work is directly related to research that aims at learning to represent visual data with a set of feature vectors without explicit supervision. Earlier attempts start from synthetic datasets with naive shape primitives, uniformly colored objects, and simple backgrounds [6, 17, 18, 19, 22, 23, 60]. They typically perform iterative inference to extract object-centric features from images, followed by a mixture-based CNN decoder applied to each slot separately for reconstruction. For example, MONet [6] and Slot Attention [60] both adopt the spatial broadcast decoder [97] to predict an RGB image and an objectness mask from each slot, and combine them via alpha masking. Recently, SLATE [83] challenges the traditional design with a Transformer-based decoder, which helps the model scale to more complex data such as textured objects and backgrounds. Such decoder has also been employed in later works [79, 85], proving its effectiveness on naturalistic inputs. Another line of works [1, 30, 81, 99] focuses on feature-level reconstruction or contrastive learning for object discovery without pixel-level decoding. However, they are not applicable for generation tasks as they cannot reconstruct images from slots. With the recent progress in 3D reconstruction, there are also methods leveraging Neural Radiance Fields (NeRF) [89, 104] and light fields [79, 86] as the slot decoder. However, they usually require multi-view images and camera poses as training data, which is incompatible with our setting. Compared to previous works, SlotDiffusion is the first attempt introducing diffusion models to unsupervised object-centric learning, achieving state-of-the-art results in both object discovery and generation.

**Unsupervised object-centric learning from videos.** Compared to images, videos provide additional information such as motion cues. Our work also builds upon recent efforts in decomposing raw videos into temporally aligned object slots [40, 46, 48, 56, 93, 106]. These works usually integrate a dynamics module to model the interactions between objects compared to their image-based counterparts. For example, STOVE [49] adopts a Graph Neural Network (GNN), while OAT [9], SAVi [46], and PARTS [110] leverage the powerful Transformer architecture. However, the limited modeling capacity of their mixture-based decoders still prevents them from scaling to more realistic videos. Some works [16, 102] thus introduce additional supervisions such as optical flow, depth maps, and initial object positions. Recently, STEVE [84] pushes the limit of fully unsupervised object-centric learning on videos with the Transformer slot decoder from SLATE and the Transformer dynamics module from SAVi. Nevertheless, the autoregressive generation mechanism of Transformer degrades its generation capacity, which we aim to address in this paper. There are also methods learning scene decomposition from videos by contrasting feature [45, 61], which cannot generate videos. Similar to SAVi and STEVE, SlotDiffusion can also be applied to video data with the Transformer-based dynamics module. In addition, we generate videos with much better temporal consistency.

**Diffusion model for generation.** Recently, diffusion models [87] (DMs) have achieved tremendous progress in generation tasks, including 2D images [11, 34, 64, 65, 71, 77], 2D videos [27, 35, 36, 82], 3D shapes [53, 58, 66], and scene editing [31, 44, 63, 75], showing their great capacity in generation quality and input conditioning. The generative process of DMs is formulated as an iterative denoising procedure with a denoising network, usually implemented as a U-Net [74]. However, the memory consumption of DMs scales quadratically with the input resolution due to the use of self-attention layers in the U-Net. To reduce the training cost, LDM [73] proposes to run the diffusion process in the latent space of a pre-trained image auto-encoder, e.g., VQ-VAE [72]. LDM also introduces a flexible conditioning mechanism via cross-attention between the U-Net feature maps and the conditional inputs. For example, recent text-guided LDMs [73] first extract text embeddings using pre-trained language models [69, 70], and then perform cross-attention between the language features and the U-Net denoising feature maps at multiple resolutions. In this work, we adopt the LDM as our slot decoder due to its strong generation capacity and low computation cost. To condition the decoder on 1D object slots, we draw an analogy with 1D text embeddings, and also apply cross-attention between slots and feature maps in the U-Net denoiser.

**Diffusion model for representation learning.** There have been a few works studying the representation learned by DMs. Baranchuk et al. [3] and Xu et al. [101] utilize the denoising feature maps of DMs for data-efficient segmentation, as the feature maps already capture object boundary information. Preechakul et al. [68] jointly trains a CNN encoder and a DM to perform auto-encoding for representation learning. DiffMAE [98] combines MAE [29] and DMs to learn better inpainting. Li et al. [52] leverages pre-trained text-guided DMs for zero-shot image classification. In contrast, SlotDiffusion does not rely on any supervision or pre-trained DMs. Instead, we incorporate it with the unsupervised Slot Attention framework to learn structured object-centric representations.

**Conditional diffusion model without labels.** Several works have explored learning conditional DMs in an unsupervised manner. Hu et al. [37] and Bao et al. [2] utilize self-supervised feature extractors to obtain image features, and then perform clustering to create pseudo labels such as cluster ids for training a conditional DM. Harvey and Wood [26] show that conditional DMs usually outperform unconditional DMs in terms of FID. Inspired by this observation, they leverage CLIP embeddings [69] of unlabeled images to train conditional DMs without manual annotations. Similarly, SlotDiffusion also learns a conditional DM without labels. Instead of relying on pre-trained feature extractors, we build upon the unsupervised Slot Attention framework, and can learn both object-centric representations and conditional DMs from scratch.

## C Details on Diffusion Models

Diffusion models are probabilistic models that learn a data distribution $p_\theta(\boldsymbol{X}_0)$ by gradually denoising a standard Gaussian distribution, in the form of $p_\theta(\boldsymbol{X}_0) = \int p_\theta(\boldsymbol{X}_{0:T}) \, d\boldsymbol{X}_{1:T}$. Here, $\boldsymbol{X}_{1:T}$ are intermediate denoising results with the same shape as the clean data $\boldsymbol{X}_0 \sim q(\boldsymbol{X})$, and $\theta$ are learnable parameters of the denoising U-Net model.

The joint distribution $q(\boldsymbol{X}_{1:T}|\boldsymbol{X}_0)$ is called the *forward process* or *diffusion process*, which is a fixed Markov Chain that gradually adds Gaussian noise to $\boldsymbol{X}_0$. The noise is controlled by a pre-defined variance schedule $\{\beta_t\}_{t=1}^T$:

$$q(\boldsymbol{X}_{1:T}|\boldsymbol{X}_0) = \prod_{t=1}^T q(\boldsymbol{X}_t|\boldsymbol{X}_{t-1}), \ \ q(\boldsymbol{X}_t|\boldsymbol{X}_{t-1}) = \mathcal{N}(\boldsymbol{X}_t|\sqrt{1-\beta_t}\boldsymbol{X}_{t-1}, \beta_t\boldsymbol{I}). \tag{8}$$

Thanks to the nice property of Gaussian distributions, $\boldsymbol{X}_t$ can be sampled directly from $\boldsymbol{X}_0$ in closed form without adding the noise $t$ times. Let $\alpha_t = 1 - \beta_t$ and $\bar{\alpha}_t = \prod_{s=1}^t \alpha_s$, we have:

$$q(\boldsymbol{X}_t|\boldsymbol{X}_0) = \mathcal{N}(\boldsymbol{X}_t|\sqrt{\bar{\alpha}_t}\boldsymbol{X}_0, (1-\bar{\alpha}_t)\boldsymbol{I}) \ \Rightarrow \ \boldsymbol{X}_t = \sqrt{\bar{\alpha}_t}\boldsymbol{X}_0 + \sqrt{1-\bar{\alpha}_t}\epsilon_t, \text{ where } \epsilon_t \sim \mathcal{N}(\boldsymbol{0}, \boldsymbol{I}). \tag{9}$$

We can now train a model to reverse this process and thus generate target data from random noise $\boldsymbol{X}_T \sim \mathcal{N}(\boldsymbol{0}, \boldsymbol{I})$. The *reverse process* $p_\theta(\boldsymbol{X}_{0:T})$ is also defined as a Markov Chain with a learned Gaussian transition:

$$p_\theta(\boldsymbol{X}_{0:T}) = p(\boldsymbol{X}_T) \prod_{t=1}^T p_\theta(\boldsymbol{X}_{t-1}|\boldsymbol{X}_t), \ \ p_\theta(\boldsymbol{X}_{t-1}|\boldsymbol{X}_t) = \mathcal{N}(\boldsymbol{X}_{t-1}|\mu_\theta(\boldsymbol{X}_t, t), \Sigma_\theta(\boldsymbol{X}_t, t)) \tag{10}$$

In practice, we do not learn the variance and usually set it to $\Sigma_t = \beta_t\boldsymbol{I}$ or $\frac{1-\bar{\alpha}_{t-1}}{1-\bar{\alpha}_t}\beta_t\boldsymbol{I}$ since it leads to unstable training [34]. Also, instead of learning the mean $\mu_\theta$ directly, we learn to predict the noise $\epsilon_t$ in Equation (8). See Ho et al. [34] for how we can sample $\boldsymbol{X}_{t-1}$ given $\boldsymbol{X}_t$ and the predicted $\epsilon_t$ at the inference stage.

The training process of diffusion models is thus straightforward given Equation (9). At each training step, we sample a batch of clean data $\boldsymbol{X}_0$ from the training set, timesteps $t$ uniformly from $\{1, ..., T\}$, and random Gaussian noise $\epsilon_t \sim \mathcal{N}(\boldsymbol{0}, \boldsymbol{I})$. We then create the noisy version of data $\boldsymbol{X}_t$ by applying Equation (9). A denoising model $\epsilon_\theta$ is trained to predict the noise with an MSE loss:

$$\mathcal{L}_{DM} = \mathbb{E}_{\boldsymbol{X}, t, \epsilon_t}[||\epsilon_t - \epsilon_\theta(\boldsymbol{X}_t, t)||^2] \tag{11}$$

## D Detailed Experimental Setup

In this section, we detail the datasets, evaluation metrics, baseline methods used in the experiments, and the implementation details of our model.

### D.1 Datasets

**CLEVRTex [43].** This dataset augments the CLEVR [42] dataset with more diverse object shapes, materials, and textures. The backgrounds in CLEVRTex also present complex textures compared to the plain gray ones in CLEVR. Therefore, this dataset is visually much more complex than CLEVR. As shown in their paper, only 3 out of 10 benchmarked unsupervised object-centric models can achieve an mIoU higher than 30%. We train our model on the training set consisting of 40k images, and test on the test set with 5k samples. We use the same data pre-processing steps, i.e. first center-crop to 192×192, and then resize to 128×128.

**CelebA [59].** This dataset contains over 200k real-world celebrity images. All images are mainly occupied by human faces, covering a large variation of poses and backgrounds. This dataset is more challenging compared to previous synthetic datasets as real-world images typically have background clutter and complicated lighting conditions. We train our model on the training set with around 160k images, and test on the test split with around 20k images. For data pre-processing, we simply resize all images to $128\times128$.

**MOVi-D/E [25].** MOVi-D and MOVi-E are the 2 most challenging versions from the MOVi benchmark generated using the Kubric simulator. Their videos feature photo-realistic backgrounds and real-world objects from the Google Scanned Objects (GSO) repository [15], where one or several objects are thrown to the ground to collide with other objects. Compared to MOVi-D, MOVi-E applies linear camera motion. We follow the official train-test split to evaluate our model. For data pre-processing, we simply resize all frames to $128\times128$.

**MOVi-Solid/Tex [84].** MOVi-Solid and MOVi-Tex share similar backgrounds and object motions with MOVi-D and MOVi-E. Besides, MOVi-Solid introduces more complex shapes, and MOVi-Tex maps richer textures to the object surfaces, making them harder to be distinguished from the background. We follow the train-test split in [84], and resize all frames to $128\times128$.

**Physion [4].** This dataset contains eight categories of videos, each showing a common physical phenomenon, such as rigid- and soft-body collisions, falling, rolling, and sliding motions. The foreground objects used in the simulation vary in categories, textures, colors, and sizes. It also uses diverse background as the scene environment, and randomize the camera pose in rendering the videos. Physion splits the videos into three sets, namely, *Training*, *Readout Fitting*, and *Testing*. We truncate all videos by 150 frames as most of the interactions end before that, and sub-sample the videos by a factor of 3 for training the dynamics model. Following the official evaluation protocol, the dynamics models are first trained on videos from the Training set under future prediction loss. Then, they observe the first 45 frames of videos in the Readout Fitting and Testing set, and perform rollout to generate future scene representations (e.g., image feature maps or object slots). A linear readout model is trained on observed and rollout scene representations from the Readout Fitting set to classify whether the two cued objects (one in red and one in yellow) contact. Finally, the classification accuracy of the trained readout model on the Testing set scene representations is reported. Please refer to their paper [4] for detailed descriptions of the VQA evaluation. For video prediction, we evaluate on the test set videos. All video frames are resized to $128\times128$ as pre-processing.

**PASCAL VOC 2012 [21].** PASCAL VOC is a real-world image dataset containing natural scenes with diverse objects. We follow Seitzer et al. [81] to train on the "trainaug" variant with 10,582 images, and test on the official validation set with 1,449 images. For data pre-processing, we resize the smaller dimension of the image to 224, and perform random cropping to $224\times224$ followed by random horizontal flip during training. For testing, we evaluate the $224\times224$ center crop of images, and ignore pixels that are unlabeled.

**Microsoft COCO 2017 [54].** Similar to VOC, MS COCO is a real-world benchmark commonly used in object detection and segmentation. We follow Seitzer et al. [81] to train on the training set with 118,287 images, and test on the validation set with 5,000 images. For data pre-processing, we resize the smaller dimension of the image to 224, and perform center crop to $224\times224$ for both training and testing. We apply random horizontal flip during training. During evaluation, we filter out crowd instance annotations, and ignore pixels where objects overlap.

### D.2 Evaluation Metrics

To evaluate the generation quality of SlotDiffusion, we compute the *mean squared error (MSE)* of the images reconstructed from the object slots. Following prior works, we scale the images to $[0, 1]$, and sum the errors over channel and spatial dimensions. As pointed out by Zhang et al. [108], MSE does not align well with human perception of visual quality as it favors over-smooth results. Therefore, we additionally compute the *VGG perceptual distance (LPIPS)* [108]. For video generation, we compute the per-frame MSE and LPIPS, and then average them over the temporal dimension. Finally, to evaluate the compositional generation results, we adopt the *Fréchet Inception Distance (FID)*[1] [32] and the *Fréchet Video Distance (FVD)*[2] [92]. We compute these metrics between 5k generated images or 500 generated videos and the entire training set of that dataset.

---

[1]Implementation from https://github.com/mseitzer/pytorch-fid.
[2]Implementation from https://github.com/universome/stylegan-v.

**Table 5:** Variations in model architectures and training settings on different datasets.

| Dataset | CLEVRTex | CelebA | MOVi-D/E | MOVi-Solid/Tex | Physion | VOC / COCO |
|---|---|---|---|---|---|---|
| Image Encoder $f_{\text{enc}}$ | ResNet18 | ResNet18 | ResNet18 | Plain CNN | ResNet18 | ViT-S/8 |
| Number of Slots $N$ | 11 | 4 | 15 | 12 | 8 | 6 / 7 |
| Slot Size $D_{\text{slot}}$ | 192 | 192 | 192 | 192 | 192 | 192 / 256 |
| Slot Attention Iteration | 3 | 3 | 2 | 2 | 2 | 3 |
| Max Learning Rate | 2e-4 | 2e-4 | 1e-4 | 1e-4 | 1e-4 | 1e-4 |
| Gradient Clipping | 1.0 | 1.0 | 0.05 | 0.05 | 0.05 | 0.05 |
| Batch Size | 64 | 64 | 32 | 32 | 48 | 64 |
| Training Epochs | 100 | 50 | 30 | 30 | 10 | 500 / 100 |

To evaluate the scene decomposition results, we compute the *FG-ARI*, *mIoU*, and *mBO* of the object masks. FG-ARI is widely used in previous object-centric learning papers, which only consider foreground objects. As suggested in [17, 43], we should also compute the mIoU which evaluates background segmentation. For mBO, it first assigns each ground-truth object mask with the max-overlapping predicted mask, and then averages the IoU of all pairs of masks. It is worth noting that on video datasets, we flatten the temporal dimension into spatial dimensions when computing the metrics, thus taking the temporal consistency of object masks into consideration, i.e., slot ID swapping of the same object will degrade the performance. Being able to consistently track all the objects is an important property of video slot models.

### D.3 Baselines

**Object-centric models.** We adopt Slot Attention [60] and SAVi [46] as representative models which use a mixture-based CNN decoder, and SLATE [83] and STEVE [84] for models with a Transformer-based decoder. We use the unconditional version of these models, i.e., they are not trained with the initial frame supervision proposed in [46]. Since we augment SlotDiffusion with a stronger ResNet18 encoder compared to the previous stacked CNN encoder, we also re-train baselines with the same ResNet18 encoder, which achieves better performance than reported in their papers. In addition, we tried scaling up the slot decoder of baselines by adding more layers and channels, but observed severe training instability beyond a certain point. Instead, SlotDiffusion shows great scalability with model sizes. Its performance grows consistently with a larger LDM U-Net. On real-world datasets PASCAL VOC and COCO, we directly copy the numbers of DINOSAUR [81] from its paper, as they did not release their code until 15 days before the paper submission deadline. For Slot Attention and SLATE here, we replace the CNN encoder with the same DINO [7] pre-trained ViT encoder. We trained longer enough and achieved better results than reported in the DINOSAUR paper.

**Video prediction and VQA.** We adopt the same baselines as SlotFormer [100]. Please refer to their papers about the implementation details of these future prediction models. We train baselines on Physion with the default hyper-parameters. For VQA, we directly copy the numbers from SlotFormer.

### D.4 Our Implementation Details

**VQ-VAE [72].** We use the same architecture for all datasets, which is adopted from LDM [73]. We use 3 encoder and decoder blocks, resulting in 4x down-sampling of the feature maps $z$ compared to input images $x$. We pre-train the VQ-VAE for 100 epochs on each dataset with a cosine learning rate schedule decaying from $1 \times 10^{-3}$, and fix it during the object-centric model training.

**SlotDiffusion.** For the object-centric model, we only replace the decoder with LDM [73] compared to Slot Attention on images and SAVi on videos. We use a modified ResNet18 encoder [46] to extract image features[3], except for real-world datasets PASCAL VOC and COCO, we use the DINO self-supervised pre-trained [7] encoder, where the ViT-S/8 variant is chosen following DINOSAUR[4] [81]. On video datasets, we train on small clips of 3 frames. For the LDM-based slot decoder, the training target is to predict the noise $\epsilon$ added to the features $z$ produced by a pre-trained VQ-VAE. Following prior works, the denoising network $\epsilon_\theta(z_t, t, \mathcal{S})$ is implemented as a U-Net [74] with global self-attention and cross-attention layers in each block. We use the same noise schedule $\{\beta_t\}_{t=1}^{T}$ and U-Net hyper-parameters as LDM [73]. The attention map from the last Slot Attention iteration is used as the predicted segmentation. See Table 5 for detailed slot configurations and training settings.

---

[3]Interestingly, we found that ResNet18 encoder leads to training divergence on MOVi-Solid and MOVi-Tex datasets for both SlotDiffusion and baselines. Therefore, we adopt the plain CNN encoder on these two datasets.

[4]Pre-trained weight obtained from the HuggingFace transformers library.

**Table 6:** Comparison of model complexity on the MOVi-D/E video datasets. We measure the training memory consumption, time per training step, and generation time of 100 videos at test stage. For training, we report the default settings (batch size 32 of length-3 video clips, frame resolution 128×128) on NVIDIA A40 GPUs. For testing, we report the inference time on NVIDIA T4 GPUs (each video contains 24 frames).

|       |              | SAVi | STEVE | SlotDiffusion |
|-------|--------------|------|-------|---------------|
| Train | Memory (GB)  | 32   | 51    | 24            |
|       | Time (s)     | 0.57 | 0.87  | 0.77          |
| Test  | Time (min)   | 0.7  | 226   | 7             |

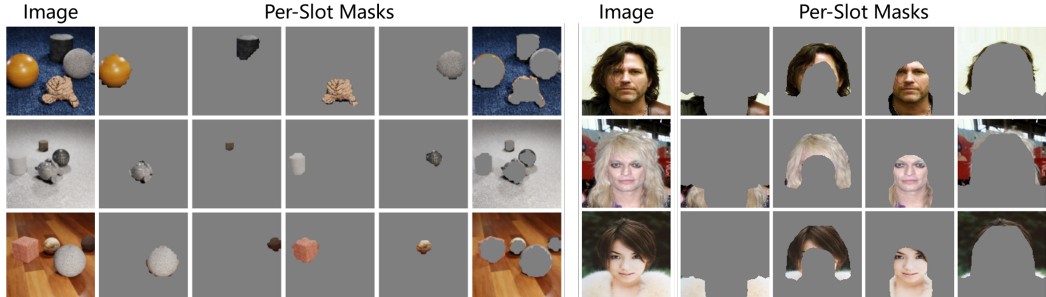

**Figure 13:** Visualization of image segmentation results on CLEVRTex (left) and CelebA (right).

**Slot-to-image decoding.** Diffusion models are notoriously slow in doing generation due to the iterative denoising process, where the diffusion step $T$ is often 1,000. Fortunately, researchers have developed several methods to accelerate this procedure. We employ the DPM-Solver [62] which reduces the sampling steps to 20. Therefore, our reconstruction speed is even faster than models with Transformer-based decoders, as they need to forward their model number of tokens (typically 1,024) times for autoregressive token prediction, while we only need 20 times.

**SlotFormer [100] on Physion.** We use its online official implementation[5]. We first pre-train a SlotDiffusion model on Physion, and use it to extract object slots from all videos and save them offline. Then, we train SlotFormer to predict slots at future timesteps conditioned on slots from input timesteps. We follow the same model architecture and training settings as [100]. For video prediction, we use the pre-trained LDM-based slot decoder to decode the predicted slots back to frames, so that we can evaluate the generation quality. For VQA, we train additional readout models over the predicted slots to answer questions. Please refer to [100] for more information.

## E Additional Experimental Results

### E.1 Computational Requirements

We empirically show the speed and GPU memory requirement at training time, as well as the time required to reconstruct 100 videos at test time of SAVi, STEVE, and SlotDiffusion in Table 6. Interestingly, our model requires the least GPU memory during training despite achieving the best performance. This is because we run the diffusion process at latent space, whose spatial dimension is only 1/4 of the input resolution. In contrast, SAVi applies CNN to directly reconstruct images at the original resolution, and STEVE uses an autoregressive Transformer to predict a long sequence of 1,024 tokens. Both of these designs consume large GPU memory. In terms of training and generation speed, SlotDiffusion ranks second. SAVi runs extremely fast since it decodes images in one-shot, while STEVE and our model both need to do iterative sampling. Thanks to the efficient DPM-Solver sampler, we only require 20 times forward pass, while STEVE requires 1,024 times.

**Total training time.** SAVi is the fastest to train which takes 30 hours on a single NVIDIA A40 GPU. STEVE and SlotDiffusion both need 20 hours to train the VAE image tokenizer. STEVE's second stage training requires 50 hours on 2 GPUs, while SlotDiffusion requires 40 hours on 1 GPU.

### E.2 Object Segmentation

**Image.** We visualize our image segmentation results in Figure 13. SlotDiffusion is able to discover objects or semantically meaningful and consistent regions from images in an unsupervised way. For

---

[5]https://github.com/pairlab/SlotFormer.

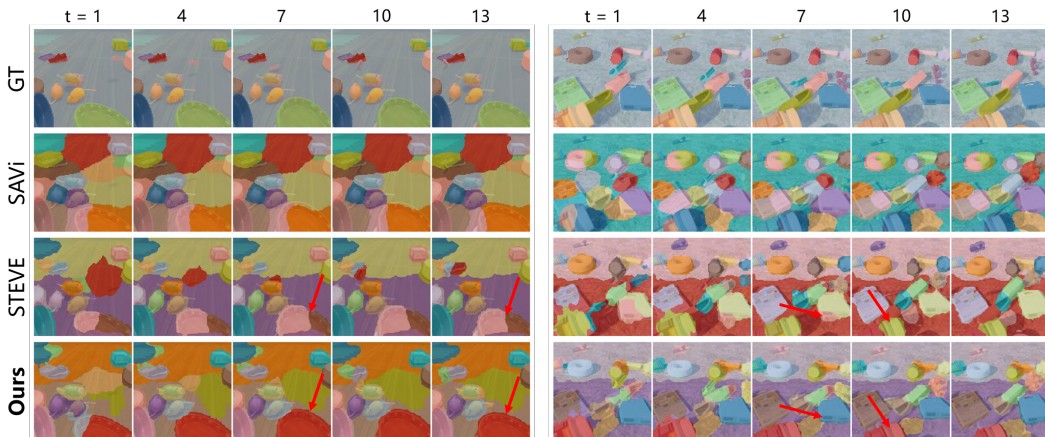

**Figure 14:** Visualization of video segmentation results on MOVi-D (left) and MOVi-E (right).

example, on CelebA, we decompose images into four parts: clothes, hairs, faces, and backgrounds. As shown in the section of compositional generation, we can generate novel samples by randomly composing these basic visual concepts. Notably, the object masks produced on CLEVRTex usually do not match the exact boundary of objects. This is because the segmentation masks are predicted in a low resolution (4x down-sampling in our case) and up-sampled to the original resolution. How to refine the object masks from a coarse initialization is an interesting future direction.

**Video.** We show more video segmentation results in Figure 14. SAVi typically generates blob-shape masks that do not follow the exact boundary of objects due to the low modeling capacity of the mixture-based decoder. Compared to STEVE, SlotDiffusion usually has more consistent tracking results and fewer object-sharing issues, especially on large objects. We also note that the shown examples of SlotDiffusion on MOVi-D have FG-ARI scores higher than 80%, although their visual quality is far from perfect. This indicates that the FG-ARI metric has been saturated, and we should focus more on better metrics such as mIoU [43].

### E.3    Compositional Generation

**Reconstruction.** As in the main paper, we first show more reconstruction results in Figure 15. Compared to baselines that produce blurry frames, SlotDiffusion is able to preserve the local appearances such as textures on the objects and backgrounds. However, there is still a large room for improvement in finer details. For example, in the top example, we cannot reconstruct the texts on the object surfaces. Also, in the bottom example, we fail to retain the smooth surface of the brown shoes.

**Generation.** In Figure 16, we compare original dataset samples to novel results produced by baselines and SlotDiffusion. Due to the limited capacity, models with the mixture-based decoder generate blurry images. Although the Transformer-based decoder helps SLATE and STEVE decompose complex scenes, they typically produce objects with distorted attributes, such as deformed object shapes and unnatural facial features. Thanks to the iterative refinement process of DMs, SlotDiffusion is able to generate high-fidelity images and videos with fine details, coherent with the real samples.

**Remark: the generation diversity of SlotDiffusion's LDM decoder.** One good property of DMs is that they can generate diverse results from a single conditional input [11, 66, 73]. This is because the conditional inputs do not fully describe the output's content. For example, in text-guided generation, the input texts usually only describe the global style and layout of the images, while LDMs have the freedom to generate diverse local details. In contrast, SlotDiffusion conditions the LDM on slots, which are designed to capture all information (position, shape, texture, etc.) about objects. Therefore, we expect our LDM to faithfully reconstruct the original objects, instead of generating diverse results, which is also a desired property in our downstream applications. For example, when performing face replacement (Figure 8 (b) in the main paper), we want to maintain the facial attributes of the source face. Nevertheless, we note that SlotDiffusion is still able to generate novel content when necessary. For example, when removing an object from the scene in image editing (Figure 8 (a) in the main paper), it inpaints the background with plausible textures.

To study how much randomness our LDM decoder has, we run it to decode the same set of slots given six different random seeds. The results are shown in Figure 17 of the submitted PDF file, where our model generates images with minor differences (mainly local textures and surface reflections), preserving the identity of objects.

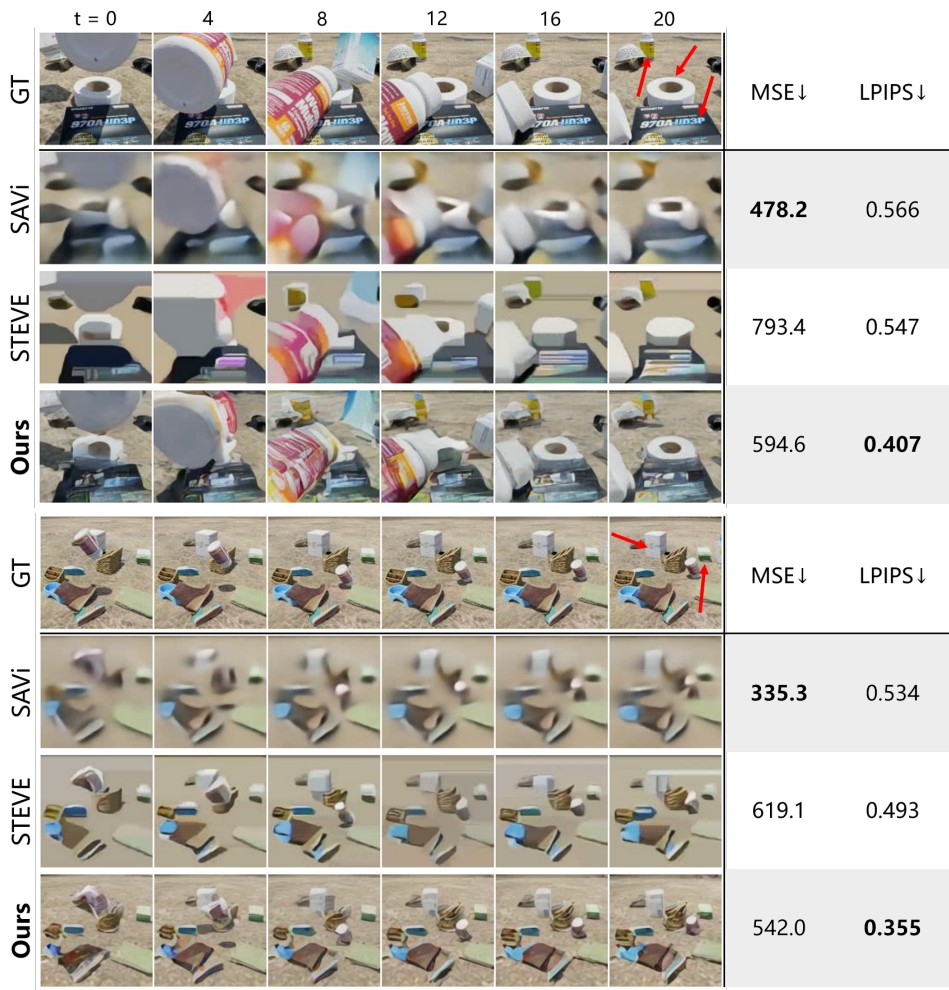

**Figure 15:** Qualitative results of slot-to-video reconstructions on MOVi-D (top) and MOVi-E (bottom).

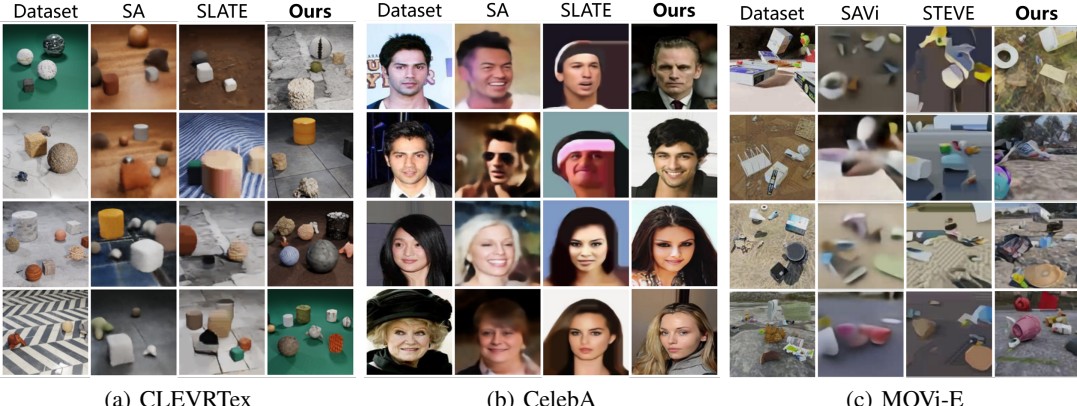

      (a) CLEVRTex                  (b) CelebA                  (c) MOVi-E

**Figure 16:** Compositional generation results on CLEVRTex (a), CelebA (b), and MOVi-E (c). Compared to baselines, SlotDiffusion produces results with higher fidelity and more details, such as textures on the objects and backgrounds, and hairs and facial features of humans, which are visually closer to the dataset samples.

## E.4 Video Prediction and VQA

We show more qualitative results of Physion video prediction in Figure 18. PredRNN produces blurry frames with objects disappearing while still achieves a low MSE, proving that MSE is a poor metric in generation tasks. Although VQFormer generates sharp images, the generated frames are almost static, lacking scene dynamics. STEVE with SlotFormer is able to predict the rough motion of objects, yet the results are of low quality. In contrast, SlotDiffusion learns highly informative slots and can decode high-fidelity images. Therefore, our method predicts videos with the correct object motion and consistent scenes, even for the challenging dynamics of deformable cloth.

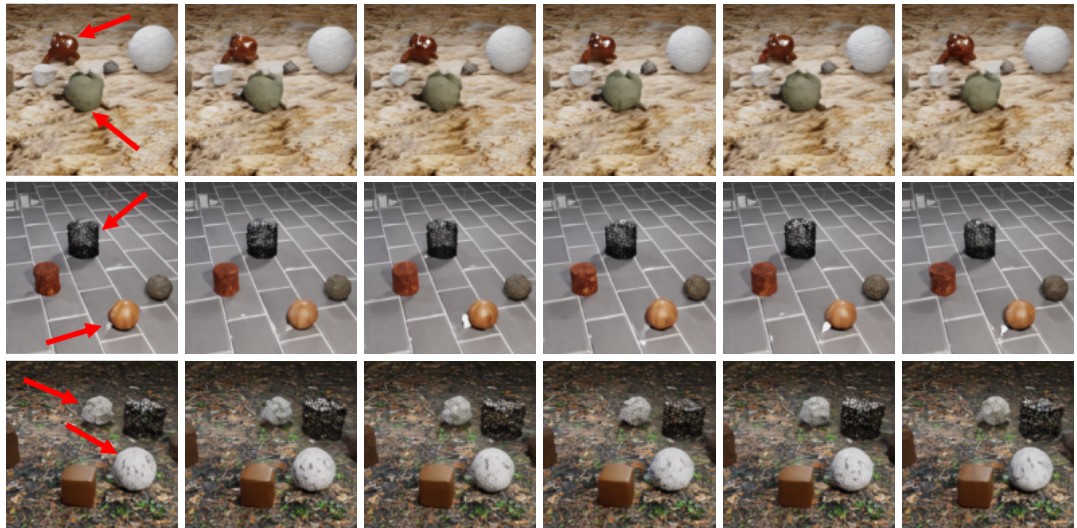

**Figure 17:** Images sampled from SlotDiffusion's LDM decoder with different random seeds.

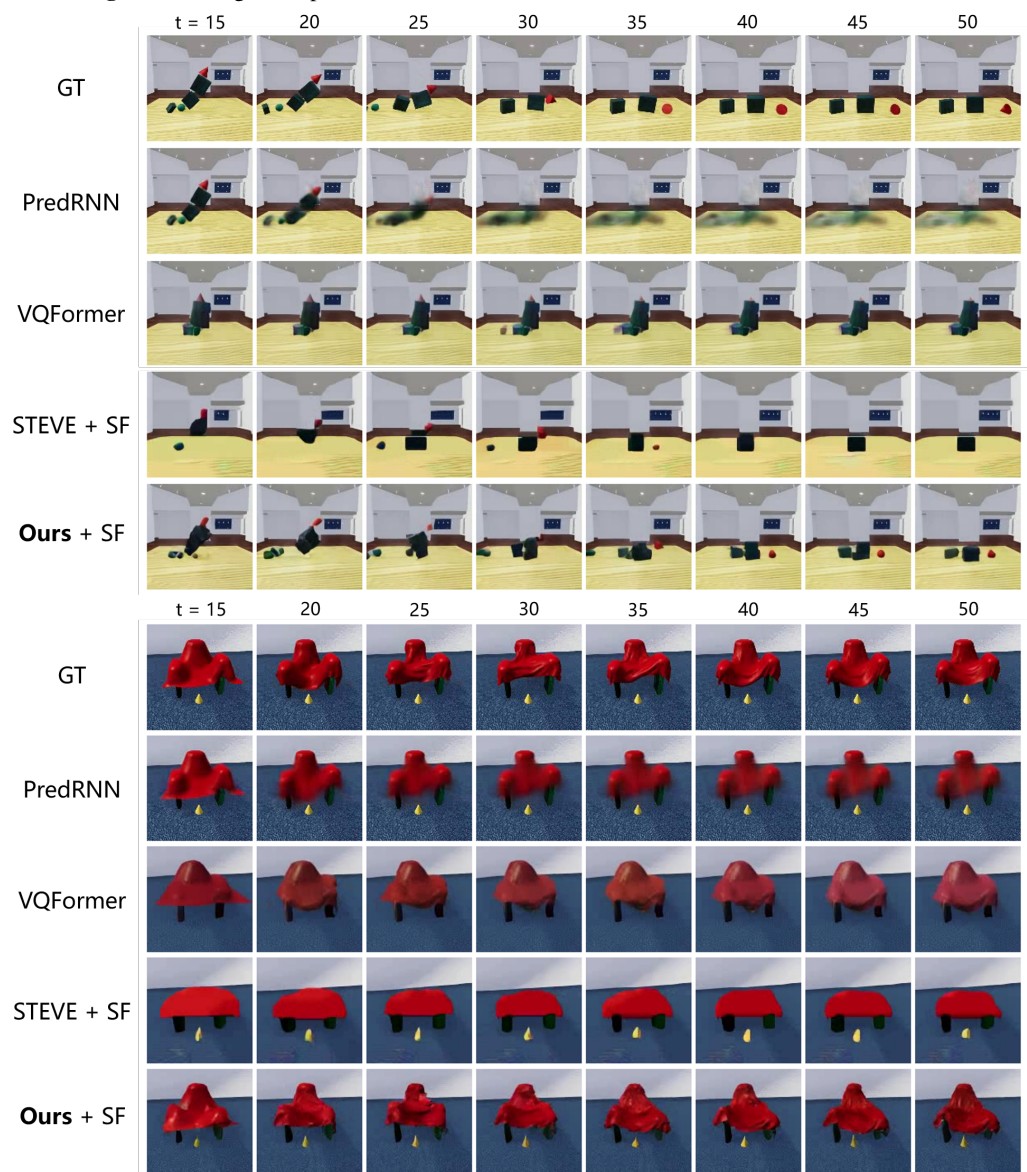

**Figure 18:** Qualitative results of video prediction on Physion. SF stands for SlotFormer.

### E.5  Failed Attempts

Here, we record some failed model variants we tried.

**Image-space diffusion model.** At the early stage of this work, we adopt an image-space DM as the slot decoder without the VQ-VAE tokenizer. This works well with low resolution inputs ($64 \times 64$). However, when we increase the resolution to $128 \times 128$, we observe color shifting artifacts which is a common issue among high resolution DMs [76, 78]. Switching to an LDM-based decoder solves this problem, as the VQ-VAE decoder always maps latent codes to images with natural color statistics. In addition, since LDM consumes less memory, we can use a larger U-Net as the denoiser, which further improves the performance.

**Prediction target of the diffusion model.** By default, our DM adopts the traditional noise prediction ($\epsilon$) target. Recent works in DM propose two other formulations, namely, data prediction ($\mathbf{x}_0$) [71] and velocity prediction ($\mathbf{v}$) [35], and claim to have better generation quality. We tried both targets in SlotDiffusion, but both model variants degenerate to the stripe pattern solution (similar to the SAVi result on MOVi-D in Figure 14) under multiple random seeds.

**Classifier-free guidance [33].** We also tried classifier-free guidance (CFG for short) to improve the generation quality of SlotDiffusion. In order to apply CFG during inference, we need to randomly drop the conditional input (i.e., slots) during model training. However, with a high dropping rate, we observe severe training instability, where the model degenerates to the stripe pattern solution. When trained with a low dropping rate, we do not observe clear improvement in the generation quality.

## F  Limitations and Future Works

**Limitations.** The ultimate goal of unsupervised object-centric models is to perform image-to-slot decomposition and slot-to-image generation on real-world data without supervision. With the help of pre-trained encoders such as DINO ViT [7], SlotDiffusion is able to segment objects from naturalistic images [21, 54]. However, the images reconstructed from slots are still of low quality, preventing unsupervised object-centric generative models from scaling to these data. In addition, objects in real-world data are not well-defined. For example, the part-whole hierarchy causes ambiguity in the segmentation of articulated objects, and the best way of decomposition depends on the downstream task. Finally, our learned slot representations are still entangled, while many tasks require disentanglement between object positions and semantics.

**Future Works.** From the view of generative modeling, we can explore better DM designs and techniques such as classifier-free guidance to improve the modeling capacity of fine local details. Moreover, our LDM decoder currently generates video frames individually without considering the temporal interactions. We can try the temporal attention layer in video DMs [27, 36] to improve the temporal consistency. To generate high-quality natural images, one solution is to incorporate pre-trained generative models such as the text-guided Stable Diffusion [73]. As discussed in the main paper, object slots in an image are similar to phrases in a sentence. We may try to map the slots to the text embedding space or via lightweight adapters [107].

From the view of object-centric learning, we want to apply SlotDiffusion to more downstream tasks such as reinforcement learning [95, 105] and 3D modeling [86, 104]. It is interesting to see if task-specific supervision signals (e.g., rewards in RL) can lead to better scene decomposition. To address the part-whole ambiguity of objects, Segment Anything [47] proposes to output three levels of masks per pixel, which cover the whole, part, and subpart hierarchy. We can apply such a design to slot-based models to support richer scene decomposition. Finally, there can be new protocols for evaluating these object-centric models. For example, MAE [29] leverages supervised fine-tuning to verify the effectiveness of their learned representations. We can also consider SlotDiffusion with labels such as class labels or object bounding boxes [46].

# G   Full Experimental Results

To facilitate future research, we report the numeric results of all bar charts used in the main paper in Table 7, Table 8, and Table 9.

**Table 7:** Full results on CLEVRTex and CelebA datasets. FG-ARI, mIoU, and mBO are given in %. $^*$ we use the masks provided by the CelebA-Mask dataset [51] to evaluate the segmentation results on CelebA. We select four classes from the provided labels, the original "hair", "cloth", and "background" classes, and a new "face" class by merging masks of all other facial attributes such as eye, nose, mouth. The data split of CelebA-Mask is slightly different from the original CelebA dataset, so we do not report this result in the main paper.

| Method | CLEVRTex | | | | | | CelebA$^*$ | | | | | |
|---|---|---|---|---|---|---|---|---|---|---|---|---|
| | FG-ARI ↑ | mIoU ↑ | mBO ↑ | MSE ↓ | LPIPS ↓ | FID ↓ | FG-ARI ↑ | mIoU ↑ | mBO ↑ | MSE ↓ | LPIPS ↓ | FID ↓ |
| SA | 67.92 | 53.70 | 57.43 | **212.4** | 0.41 | 134.57 | 32.13 | 26.26 | 28.79 | **243.3** | 0.28 | 89.63 |
| SLATE | 67.50 | 56.82 | 59.35 | 313.3 | 0.39 | 88.73 | 40.23 | 35.72 | 37.44 | 744.8 | 0.32 | 78.95 |
| **Ours** | **68.39** | **57.56** | **61.94** | 237.5 | **0.13** | **32.07** | **44.01** | **38.46** | **41.35** | 439.0 | **0.21** | **27.72** |

**Table 8:** Full results on MOVi-D and MOVi-E datasets. FG-ARI, mIoU, and mBO are given in %.

| Method | MOVi-D | | | | | | MOVi-E | | | | | |
|---|---|---|---|---|---|---|---|---|---|---|---|---|
| | FG-ARI ↑ | mIoU ↑ | mBO ↑ | MSE ↓ | LPIPS ↓ | FVD ↓ | FG-ARI ↑ | mIoU ↑ | mBO ↑ | MSE ↓ | LPIPS ↓ | FVD ↓ |
| SAVi | 39.63 | 17.76 | 18.49 | **237.2** | 0.53 | 1123.8 | 46.70 | 23.49 | 22.23 | **255.2** | 0.55 | 1110.2 |
| STEVE | 49.43 | 29.51 | 29.16 | 673.2 | 0.51 | 857.0 | 56.04 | 27.86 | 27.97 | 603.2 | 0.51 | 796.4 |
| **Ours** | **52.09** | **31.65** | **31.24** | 632.7 | **0.40** | **493.2** | **59.99** | **30.16** | **30.22** | 586.5 | **0.39** | **353.3** |

**Table 9:** Full results on MOVi-Solid and MOVi-Tex datasets. FG-ARI, mIoU, and mBO are given in %.

| Method | MOVi-Solid | | | | | | MOVi-Tex | | | | | |
|---|---|---|---|---|---|---|---|---|---|---|---|---|
| | FG-ARI ↑ | mIoU ↑ | mBO ↑ | MSE ↓ | LPIPS ↓ | FVD ↓ | FG-ARI ↑ | mIoU ↑ | mBO ↑ | MSE ↓ | LPIPS ↓ | FVD ↓ |
| SAVi | 44.95 | 30.74 | 29.88 | **158.1** | 0.44 | 1752.1 | 26.40 | 7.60 | 7.01 | **193.1** | 0.40 | 1139.2 |
| STEVE | 60.16 | 40.72 | 42.67 | 525.7 | 0.45 | 1245.2 | 39.02 | 34.27 | 34.89 | 476.6 | 0.32 | 704.6 |
| **Ours** | **69.13** | **46.55** | **48.68** | 240.3 | **0.17** | **361.9** | **42.32** | **38.33** | **39.03** | 208.0 | **0.11** | **242.2** |

