# OpenReview forum: "SlotDiffusion: Object-Centric Generative Modeling with Diffusion Models"
_NeurIPS.cc/2023/Conference — NeurIPS 2023 spotlight_

### Official Review · Reviewer_JU9y · 2023-07-03

**Soundness:** 3 good
**Presentation:** 3 good
**Contribution:** 2 fair
**Rating:** 6
**Confidence:** 5

**Summary:**

This paper presents SlotDiffusion, an object-centric Latent Diffusion Model (LDM) designed for image and video data. The authors focus on improving slot-to-image decoding, a critical aspect of visual generation. SlotDiffusion outperforms previous slot models in unsupervised object segmentation and visual generation across multiple datasets. The model's learned object features also enhance video prediction quality and temporal reasoning tasks. The scalability of SlotDiffusion is demonstrated on real-world datasets like PASCAL VOC and COCO, showcasing its potential when combined with self-supervised pre-trained image encoders.

**Strengths:**

- This direction of slot-based generation is promising.
- The experimental section provides fruitful results that effectively validate the proposed method from various perspectives.

**Weaknesses:**

- In the experimental section, the authors primarily compare their method with mixture and transformer-based baselines. As a result, the main contribution of the observed gains may primarily stem from the diffusion models themselves, potentially undermining the novelty of the proposed methods.
- It would be beneficial to evaluate the proposed method on the CelebaMask dataset, as mentioned in Line 235.
- In Line 96, the authors mention the differences between their work and Jiang et al. It would be useful if they also included a basic comparison in the experimental results.
- Regarding Figure 11, it would be helpful to understand the performance gap between your method and the baseline at different diffusion steps.
- The paper lacks additional discussion. Since slot-attention is a paradigm for identifying meaningful visual concepts, there are other self-supervised approaches, such as [1], [2], and [3], that achieve similar goals. Including a discussion relating to these approaches would be appreciated. It seems that [1] follows a similar process of obtaining semantic meaning in a post-hoc manner (e.g., sampling, observation, assigning semantic meaning).

[1]. Self-Guided Diffusion Models
[2].Why Are Conditional Generative Models Better Than Unconditional Ones?
[3].Visual Chain-of-Thought Diffusion Models

**Questions:**

As above.

I am inclined to raise my score if my concerns are fully resolved.

**Limitations:**

yes

---

> ### Author Rebuttal · Authors · 2023-08-08
>
> Thank you for your review and insightful comments.
>
>
> **1. Are performance gains mainly due to the use of diffusion models?**
>
> A: Yes, the performance gains indeed mainly come from the use of an LDM-based slot decoder, as diffusion models have both strong modeling capacity and generation quality. We actually view the introduction of diffusion models to object-centric learning as one of our major contributions driving the performance gains. It is true that LDMs and Slot Attention were known individually. But the fact that incorporating LDMs with slot conditioning can improve both unsupervised segmentation and visual generation was neither known nor obvious before our work. We believe that the breadth of our experiments and the strong results justify this design choice. We would also like to note that both reviewer GFM9 and reviewer jjQP recognize our diffusion-based slot decoder as novel and sound, which is a contribution to object-centric learning.
>
> Besides the general idea, we also made several technical contributions:
> - We study how to condition the LDM on slots to guide the denoising process. Drawing inspiration from text-guided generation, we propose to fuse U-Net feature maps and slots via cross-attention. As shown in our experiments, this conditioning mechanism facilitates both object segmentation and visual generation;
> - SlotDiffusion is also a novel conditional DM trained in a fully unsupervised manner (see extended discussions in the response to `5.`). Without labels or pre-trained feature extractors (except on VOC and COCO), our method learns useful object-centric features from scratch. Thanks to the slot-based representation, we can perform image editing and compositional generation in a flexible and interpretable way.
>
>
> **2. Evaluation on the CelebA-Mask dataset [1].**
>
> A: Thanks for bringing up this dataset. We select four classes from the provided labels, the original “*hair*”, “*cloth*”, and “*background*” classes, and a new “*face*” class by merging masks of all other facial attributes such as eye, nose, mouth. The segmentation results of SlotDiffusion and baselines are shown below. Our method consistently outperforms baselines in all metrics.
>
> |CelebA-Mask|FG-ARI &uarr;|mIoU &uarr;|mBO &uarr;|
> |-|-|-|-|
> |Slot Attention|32.13|26.26|28.79|
> |SLATE|40.23|35.72|37.44|
> |**SlotDiffusion**|**44.01**|**38.46**|**41.35**|
>
> [1] Lee, Cheng-Han, et al. "Maskgan: Towards diverse and interactive facial image manipulation." CVPR. 2020.
>
>
> **3. Basic comparison with LSD [2].**
>
> A: Thanks for this great suggestion. Since LSD has not released their code, we can only compare with it on CLEVRTex and MOVi-E where we both conduct experiments. Also, for fair comparison, we adopt the same pre-trained VAE as LSD does. LSD treats MOVi-E as an image dataset and trains the model on sub-sampled video frames. We follow this protocol and train a SlotDiffusion model on the same video frames.
>
> The results are summarized in **Table 2 of the submitted PDF file**. On CLEVRTex, we achieve better FG-ARI, and comparable results in mIoU, mBO, and FID. On MOVi-E, we outperform LSD in both segmentation and generation metrics. However, since LSD’s code is not yet publicly available, there could be minor differences in the implementation details (parameter size, data pre-processing, etc.). Thus, our numbers can only serve as a preliminary comparison.
>
> [2] Jiang, Jindong, et al. "Object-centric slot diffusion." arXiv preprint arXiv:2303.10834 (2023).
>
>
> **4. Performance gap between SlotDiffusion and baselines at different diffusion steps $T$.**
>
> A: We plot the best baseline results on each dataset as dashed lines in **Figure 2 of the submitted PDF file** to compare with our method using different $T$. We do not plot MSE as it favors blurry results which does not align with human perception. SlotDiffusion outperforms baselines in LPIPS, FG-ARI, and mIoU across different $T$, demonstrating robustness against this hyper-parameter.
>
>
> **5. Additional discussions about related works.**
>
> A: We thank the reviewer for pointing us to these papers. They are absolutely relevant as they also learn conditional diffusion models on datasets without any labels. We will include them in the Related Work section in the camera-ready version of the paper. Here is our extended discussions relating to these works:
>
> > **Conditional diffusion model without labels**. Several works have explored learning conditional DMs in an unsupervised manner. Hu et al. [3] and Bao et al. [4] utilize self-supervised feature extractors to obtain image features, and then perform clustering to create pseudo labels for training a conditional DM. Harvey et al. [5] show that conditional DMs usually outperform unconditional DMs in terms of FID. Inspired by this, they leverage CLIP embeddings [6] of unlabeled images to train conditional DMs without manual annotations. Similarly, SlotDiffusion also learns a conditional DM without labels. Instead of relying on pre-trained feature extractors, we build upon the unsupervised Slot Attention framework, and can learn both object-centric representations and conditional DMs from scratch.
>
> [3] Hu, V. T., et al. "Self-Guided Diffusion Models." CVPR. 2023.
>
> [4] Bao, Fan, et al. "Why Are Conditional Generative Models Better Than Unconditional Ones?." arXiv preprint arXiv:2212.00362 (2022).
>
> [5] Harvey, William, and Frank Wood. "Visual Chain-of-Thought Diffusion Models." arXiv preprint arXiv:2303.16187 (2023).
>
> [6] Radford, Alec, et al. "Learning transferable visual models from natural language supervision." ICML. 2021.

---

> > ### Comment · Reviewer_JU9y · 2023-08-18
> >
> > Thanks to the author's effort, I have read the other reviewer's opinion, and I feel that my concerns have been resolved with the extra experiment and clarification. Therefore, I have raised my score.

---

> > > ### Author Response · Authors · 2023-08-18
> > > **Thank you!**
> > >
> > > Thank you for updating your review! We are happy to hear that we were able to address your concerns. We will include these results in the final version of the paper.

---

### Official Review · Reviewer_jjQP · 2023-07-04

**Soundness:** 3 good
**Presentation:** 4 excellent
**Contribution:** 3 good
**Rating:** 6
**Confidence:** 4

**Summary:**

The paper presents, SlotDiffusion, an object-centric model that incorporates a latent diffusion model as an image decoder. Extracted slots are used as a conditioning signal for the cross-attention layers of the diffusion UNet. Other than that, the model follows the formulation of SA/STEVE/SAVi models. SlotDiffusion is tested on segmentation, reconstruction and compositional generation tasks on various simulated and real datasets, comparing favourably against Slot Attention, SLATE, STEVE, SAVi and DINOSAUR models in their respective settings. The paper additionally considers the tasks of video prediction and VQA by augmenting the model with SlotFormer, where similarly strong results are observed.

**Strengths:**

The proposal incorporates a latent diffusion model as a decoder, which is novel in the context of object-centric learning (up to concurrent work).

The model shows strong results on a variety of benchmarks.

The writing is clear and easy to follow, and the presentation is good with only a few missing details (see weaknesses/questions).

**Weaknesses:**


While the method shows improved reconstruction quality and segmentation results over methods that do not use LDM as a decoder (and are otherwise, similar or identical), it is not clear if the LDM Unet model is comparable to a small spatial broadcast decoder (as in SA) or transformer decoder of SLATE/STEVE. Are these model components comparable in terms of e.g. the number of parameters? As stated in Appendix D.3 (L180) "We found that using a _larger decoder_ and training longer leads to better performance". Could this be the reason for the improved results?

Additionally, there are some more recent works [Jie et al., Singh et al.] omitted, which might serve as better points of comparison than SA/SAVi/SLATE/STEVE.

It is not entirely clear what procedure is adapted for video prediction and VQA tasks, where the SlotDiffusion is augmented with SlotFormer. Would this not remove the LDM entirely from the model?


Singh et al. "Neural Systematic Binder"

Jie et al. "Improving Object-centric Learning with Query Optimization"

**Questions:**

For evaluating the segmentation, what masks are used? Is this the attention of the final slot attention iteration? It would be helpful to clarify how the masks are predicted for evaluation.

How much do samples from the decoder vary? In other words, can the LDM model maintain the ability to generate diverse samples? Or does initial noise input have only limited to no effect?

In the checklist, it is mentioned that results are reported as an average of 3 seeds and the std. dev. are small. Would it be possible to report them, at least for SlotDiffusion models? It is particularly interesting given the added stochasticity of the LDM.

In Figure 2, are the 3 models with different decoders specific models, eg SA, SLATE and SlotDiffusion?


**Limitations:**


The limitations seem to be largely addressed. Perhaps it would be worth mentioning that it appears that the learned slots encode the position of the objects they represent. This means that moving an object in a scene would require having a slot for the same object in a new position.

---

> ### Author Rebuttal · Authors · 2023-08-08
>
> We thank the reviewer for the constructive comments. We are glad to see the positive assessment of our paper and appreciate the detailed feedback.
>
> **1. Parameter & size comparison of SlotDiffusion and baselines.**
>
> A: We list the parameter size of SlotDiffusion and baselines on the MOVi-E dataset below. Our model is indeed larger than SAVi and STEVE. We tried scaling up these baselines by adding more layers and channels, but observed severe training instability beyond a certain point. Current sizes yielded optimal results. Instead, SlotDiffusion shows great scalability with model sizes. Its performance grows consistently with a larger LDM U-Net. Such scalability is a good property observed in recent diffusion model works, e.g. Stable Diffusion.
>
> We also note that despite a larger model size, the memory requirement of SlotDiffusion during training is lower than both baselines, and the training and inference speed of SlotDiffusion is faster than STEVE, as reported in Appendix E.1.
>
> |Model|SAVi|STEVE|SlotDiffusion|
> |-|-|-|-|
> |Parameter Size (MB)|28|63|110|
>
> **2. Comparison with more recent baselines.**
>
> A: Thanks for bringing up these related works. Below, we compare them to SlotDiffusion on CLEVRTex.
>
> **SysBinder** [1] focuses on disentangling visual concepts from images. Its segmentation and generation performance is similar to SLATE as shown in their paper. We trained SysBinder and report its results below. SysBinder achieves slightly better segmentation results than SLATE, while its generation quality is worse. In contrast, SlotDiffusion consistently outperforms SysBinder across all metrics.
>
> |CLEVRTex|FG-ARI &uarr;|mIoU &uarr;|mBO &uarr;|MSE &darr;|LPIPS &darr;|FID &darr;|
> |-|-|-|-|-|-|-|
> |SLATE|67.50|56.82|59.35|313.3|0.39|88.73|
> |SysBinder|67.80|57.01|60.55|322.3|0.39|90.73|
> |**SlotDiffusion**|**68.39**|**57.56**|**61.94**|**237.5**|**0.13**|**32.07**|
>
> **BO-QSA** [2] applies bi-level optimization in the iterative slot update process. This idea is orthogonal to our modifications to the slot decoder, and thus can also be used in SlotDiffusion (dubbed *BO-SlotDiffusion*). The comparison between BO-QSA and our methods is shown below. BO-SlotDiffusion achieves comparable segmentation results with BO-QSA. In addition, both SlotDiffusion and BO-SlotDiffusion outperform BO-QSA by sizable margins in generation quality. This proves that SlotDiffusion can benefit from recent advances in slot-based methods.
>
> |CLEVRTex|FG-ARI &uarr;|mIoU &uarr;|mBO &uarr;|MSE &darr;|LPIPS &darr;|FID &darr;|
> |-|-|-|-|-|-|-|
> |BO-QSA|**80.47**|64.26|66.27|268.2|0.38|98.93|
> |SlotDiffusion|68.39|57.56|61.94|237.5|0.13|32.07|
> |BO-SlotDiffusion|78.50|64.35|**68.68**|240.3|0.13|**30.06**|
>
> [1] Singh, Gautam, et al. "Neural systematic binder." ICLR. 2023.
>
> [2] Jia, Baoxiong, et al. "Improving object-centric learning with query optimization." ICLR. 2023.
>
> **3. Clarity on SlotDiffusion with SlotFormer for video prediction and VQA on Physion.**
>
> A: We will improve clarity by incorporating the following information: "*We first pre-train SlotDiffusion on Physion videos, and use it to extract slots from video frames and save them offline. Then, we train SlotFormer by taking in slots from previous timesteps, and predicting slots at future timesteps.*"
>
> In VQA, we train additional readout models over the predicted future slots to answer questions. As you correctly assume, the LDM decoder is not involved here. We mainly want to test whether SlotDiffusion can learn informative slots that are useful for downstream tasks.
>
> In video prediction, we leverage LDM to decode the predicted future slots back to video frames, so that we can evaluate the generation quality. The better video prediction results prove the effectiveness of our LDM-based slot decoder.
>
> **4. Segmentation mask output of SlotDiffusion.**
>
> A: Yes, the attention map of the final slot attention iteration is used as the predicted segmentation mask. We will clarify this in the revised manuscript.
>
> **5. The generation diversity of SlotDiffusion’s LDM decoder.**
>
> A: This is a good question. Previous LDMs show diversity in conditional generation because the conditional inputs underspecify the output’s content. For example, in text-guided generation, the input text usually only describes the global style and layout of the images. This leaves LDMs with the freedom to generate diverse local details.
>
> In contrast, SlotDiffusion conditions the LDM on slots, which are designed to capture all information about objects (position, shape, texture, etc.). Therefore, we expect our LDM to faithfully reconstruct the original objects, instead of generating diverse results. This is also a desired property in our applications. For example, when performing face replacement (Figure 8 (b) of the main paper), we want to maintain the facial attributes of the source face.
>
> To study how much the initial noise map affects the generation results of our LDM decoder, we run it to decode the same set of slots extracted from CLEVRTex images given different random seeds. The results are shown in **Figure 1 of the submitted PDF file**. Our LDM generates images with minor differences, preserving the identity of objects.
>
> Nevertheless, we note that SlotDiffusion is still able to generate novel content when necessary. For example, when removing an object from the scene in image editing (Figure 8 (a)), it inpaints the background with plausible textures.
>
> **6. Std. dev. of SlotDiffusion results.**
>
> A: We have reported the experimental results of SlotDiffusion with mean and std. dev. in **Table 1 of the submitted PDF file**. Overall, the std. dev. is relatively small, usually less than 2\% of the mean.
>
> **7. Specific models in Figure 2.**
>
> A: Yes. Mixture, Transformer, and Diffusion decoders stand for SA/SAVi, SLATE/STEVE, and SlotDiffusion models, respectively. We will clarify this in the caption.
>
> **8. Positional bias of the learned object slots.**
>
> A: Thanks. We will add this to the Limitation Section.

---

> > ### Comment · Reviewer_jjQP · 2023-08-10
> > **Rebuttal Acknowledgment**
> >
> > I thank the Authors for their thorough and detailed response. I believe the majority of my concerns raised in the initial review have been addressed and I would encourage the authors to include the clarifications in the paper where possible. Additional results with more recent methods are encouraging as well.
> >
> > Finally, I would also like to suggest including the discussion above about scaling the decoders from prior work as well. Considering that incorporating LDM decoder is one of the core ideas presented in this paper, it strengthens the message to show better scaling behaviour than alternatives from prior work.
> >
> > I currently have no further questions but will follow any discussion that might develop and make changes to my recommendation afterwards.

---

> > > ### Author Response · Authors · 2023-08-11
> > > **Re: Rebuttal Acknowledgment**
> > >
> > > We also want to thank the reviewer again for pointing out this confusion and for all the other insightful comments. They really help us improve our work! Please let us know if you have any further questions in the future.

---

> > > > ### Comment · Reviewer_jjQP · 2023-08-18
> > > > **Rating update**
> > > >
> > > > As stated before, the paper is solid and the additional discussion and answers provided by the authors strengthened the arguments, so just letting authors know that I have updated rating as the discussion period is drawing to the end.

---

> > > > > ### Author Response · Authors · 2023-08-18
> > > > > **Thank you!**
> > > > >
> > > > > Thank you for updating your rating! We will include these results in the final version of the paper.

---

### Official Review · Reviewer_GFM9 · 2023-07-06

**Soundness:** 4 excellent
**Presentation:** 4 excellent
**Contribution:** 3 good
**Rating:** 7
**Confidence:** 4

**Summary:**

A latent diffusion-based slot decoder is introduced to improve the segmentation and reconstruction quality for slot-based autoencoders. This decoder addresses issues with previous advanced Transformer-based decoders such as SAVi and STEVE which i) do not preserve the spatial dimension of images and ii) lack an iterative refinement component for capturing high-frequency image details. SlotDiffusion combines a pretrained VQ-VAE with a cross-attention denoising diffusion decoder. The key idea is to treat slots like text embeddings and therefore to use cross-attention in the decoder following text-guided latent diffusion models. Rigorous qualitative and quantitative results confirm the improvements in segmentation and reconstruction quality.

**Strengths:**

#### Originality

- This work fills a gap in the literature by exploring how to combine ideas from latent diffusion modeling with slot-based object-centric learning.
- The insight that slots can be used like text-embeddings in the cross-attention module of a denoising diffusion decoder is novel and potentially quite powerful.

#### Quality

- The technical description of the model is sound and the experiments are rigorous (2 complex image datasets, four video datasets, 1 video prediction and reasoning benchmark, and 2 real-world image datasets).

#### Clarity

- The paper is well-written and easy to follow.

#### Significance

- I believe this work is a valuable contribution to the object-centric learning community.
- In particular, the breadth of experiments in this paper help demonstrate the value of this research direction and the generality of the models being explored.

**Weaknesses:**

I have very few concerns with this work.

- While I applaud the authors for evaluating SlotDiffusion on video prediction, VQA, and real-world image datasets, the results on these benchmarks are not as impressive as on the synthetic datasets.
- Moreover, scaling these models to real-world images appears to require a complex, multi-stage modeling effort with various important hyperparameters to be tuned: 1) pretraining the DINO ViT model, then 2) pretraining the VQ-VAE model, then 3) stitching all these together and training the denoising diffusion slot decoder.
- These are not big issues, but it suggests that perhaps something fundamental is being missed by this approach.

**Questions:**

- Table 2: What is the improvement with respect to? What is the Obs. % and Dyn. % for random guessing?

---

Update after rebuttal: I am maintaining my initially positive outlook on this paper.

**Limitations:**

Limitations and Broader Impacts are sufficiently discussed in Appendix F and G. I would encourage the authors to include this in the main text, space permitting.

---

> ### Author Rebuttal · Authors · 2023-08-08
>
> We thank the reviewer for the positive feedback and valuable comments.
>
>
> **1. Scaling to real-world data requires complex efforts and designs.**
>
> A: We would like to note that the weights of DINO ViT and the architecture of VQ-VAE are directly adopted from prior works with minimal modifications. Also, SlotDiffusion turns out to be robust to hyper-parameter choices in these building blocks.
>
> That being said, we completely agree with the reviewer that it is still challenging to scale up current unsupervised slot-based methods to real-world data, due to the variations of objects and backgrounds in real-world images. We have discussed the limitations of this work as well as recent object-centric models in Appendix F, which indeed poses many future directions to explore.
>
>
> **2. Improvement with respect to random guessing in Table 2 (VQA task on Physion).**
>
> A: We notice that there seems to be an error in the question “*What is the improvement with respect to?*”, and we are not entirely sure which baseline the reviewer is referring to. Based on the question that follows, we assume the reviewer is asking about the "random guessing baseline”.
>
> The questions in the Physion dataset are binary classification questions (whether the two objects contact or not). Therefore, the accuracy for random guessing is 50%. As a comparison, the Obs. and Dyn. accuracy of our method is 67.5% and 69.7%, respectively.
>
> Please let us know if this is the correct interpretation or if there is any specific baseline you would like us to compare with.
>
>
> **3. Moving the Limitations and Broader Impacts sections to the main text.**
>
> A: Thanks for this great suggestion. We will include them in the main text in the camera-ready version of the paper, as we will be allowed one additional content page then.

---

> > ### Comment · Reviewer_GFM9 · 2023-08-13
> > **Thanks for the response**
> >
> > The authors have provided answers to my questions and comments in a satisfactory manner. As to the question about Table 2, I do believe I was asking about the performance of random guessing! Thanks.

---

> > > ### Author Response · Authors · 2023-08-13
> > > **Re: Thanks for the response**
> > >
> > > We also want to thank the reviewer again for the questions. They help us improve our work and shed light on future directions.

---

### Official Review · Reviewer_QakT · 2023-07-07

**Soundness:** 3 good
**Presentation:** 4 excellent
**Contribution:** 3 good
**Rating:** 7
**Confidence:** 4

**Summary:**

The paper proposes a diffusion-based decoder for slot-centric models. Previously slot-centric models are trained with spatial broadcast or transformer autoregressive decoders, however these decoders result in a comparitively poor image generation or segmentation. The paper shows that using diffusion based decoders can result in better generation, segmentation and video prediction.

**Strengths:**

i) The paper does a dense evaluation of it's method
ii) The paper gets better peformance on almost all of the settings
iii) The paper is well written and is easy to understand.


**Weaknesses:**

I couldn't find any major weakness in the paper.
Although I have a few questions that i have mentioned below.

**Questions:**

i) In Table 1, how does the model compare against spatial broadcast decoder, instead of steve that uses transformer decoder?

ii) On real world benchmarks how would this method compare against CutLER (https://arxiv.org/abs/2301.11320), a method that doesn't use any slot-based training but does normalized cuts on top of Dino features and calculates segmentation from there.

iii) How does the method compare on original CLEVR, as i find all the fg-ari scores in Figure 2 to be below 0.7.

iv) What about classifier-free guidance? have the authors tried that with slots?

v) I would be interested in seeing (maybe in future) how does this method do when coupled with supervised labels, like evaluations shown in classical SSL methods such as MAE (https://arxiv.org/abs/2111.06377) or recent Slot-based methods such as Slot-TTA (https://arxiv.org/abs/2203.11194)



**Limitations:**

Yes

---

> ### Author Rebuttal · Authors · 2023-08-08
>
> We thank the reviewer for their detailed feedback and encouraging comments.
>
>
> **1. Performance of spatial broadcast decoder based models in Table 1 (video prediction on Physion).**
>
> A: We train SlotFormer over slots extracted by SAVi, and report its video prediction results below. SlotDiffusion + SlotFormer outperforms SAVi + SlotFormer by a sizable margin in all metrics.
>
> |Physion Video Prediction|MSE &darr;|LPIPS &darr;|FVD &darr;|
> |-|-|-|-|
> |SAVi + SlotFormer|746.3|0.62|1349.4|
> |STEVE + SlotFormer|832.0|0.43|930.6|
> |**SlotDiffusion** + SlotFormer|**477.7**|**0.26**|**582.2**|
>
>
> **2. Comparison with CutLER [1] on unsupervised object segmentation.**
>
> A: We ran additional experiments to compare with CutLER on *COCO val2017* and summarize the results below. *Prev. SOTA* is the previous state-of-the-art results reported in the CutLER paper. SlotDiffusion outperforms *Prev. SOTA*, but is worse than CutLER. This is because CutLER involves multi-round self-training of complicated object segmentation models (Cascade Mask R-CNN [2]), while our method leverages simple Slot Attention for object discovery. An interesting future direction is to study how to combine slot-based models with well-studied object segmentation approaches to further boost performance.
>
> |COCO val2017|AP$_{50}^{\text{mask}}$ &uarr;|AP$_{75}^{\text{mask}}$ &uarr;|AP$^{\text{mask}}$ &uarr;|
> |-|-|-|-|
> |Prev. SOTA|9.4|3.3|4.3|
> |CutLER|**18.9**|**9.7**|**9.2**|
> |**SlotDiffusion** + DINO ViT|13.2|6.0|6.2|
>
>
> [1] Wang, Xudong, et al. "Cut and learn for unsupervised object detection and instance segmentation." CVPR. 2023.
>
> [2] Cai, Zhaowei, and Nuno Vasconcelos. "Cascade r-cnn: Delving into high quality object detection." CVPR. 2018.
>
>
> **3. Model performance on the original CLEVR dataset.**
>
> A: We did not experiment on CLEVR in our initial submission since its images are visually very simple. We trained SlotDiffusion on the CLEVR (with masks) dataset, and report the results below. SlotDiffusion outperforms SLATE, but is worse than Slot Attention. This is because Slot Attention decodes object masks with the same resolution as input images, while we use the cross-attention maps from the final slot attention iteration as the segmentation masks. Its 4x downsampled resolution limits the performance of our method. This result is also consistent with concurrent work [3] (see their Appendix B). Nevertheless, we achieve an FG-ARI of over 90%, which is still reasonably high.
>
> We note that despite a better performance on simple datasets, directly decoding masks at the original resolution prevents Slot Attention and SAVi from scaling to complicated datasets, as shown in our experiments. This is because using raw pixels as reconstruction targets biases the model towards low-level appearance features such as color statistics, which works well on CLEVR with uniformly colored objects and plain gray backgrounds. However, to decompose scenes with textured objects and backgrounds, the model needs to learn semantic features, which requires high-level reconstruction targets such as image features tokenized by pre-trained VAEs.
>
> |CLEVR|FG-ARI &uarr;|mIoU &uarr;|mBO &uarr;|
> |-|-|-|-|
> |Slot Attention|**95.67**|**76.23**|**78.67**|
> |SLATE|88.56|66.62|67.42|
> |**SlotDiffusion**|90.32|71.63|72.05|
>
> [3] Jiang, Jindong, et al. "Object-centric slot diffusion." arXiv preprint arXiv:2303.10834 (2023).
>
>
> **4. Classifier-free guidance (CFG for short) with SlotDiffusion.**
>
> A: Thanks for this great suggestion. We implemented SlotDiffusion with CFG in preparation of this rebuttal. In order to apply CFG at inference time, it is necessary to randomly drop the conditional input (i.e. slots) during training. We observed severe training instability when using a high dropping rate, where the segmentation degenerated to stripe patterns (as in the SAVi results in Figure 3 of the main paper). When training with a low dropping rate, we did not observe clear improvements in the testing generation results.
>
> We will add CFG to Appendix E.5 as one of our failed attempts and Appendix F as an interesting future direction.
>
>
> **5. SlotDiffusion coupled with supervised labels or recent slot-based methods such as Slot-TTA [4].**
>
> A: We agree that combining SlotDiffusion with supervised labels is an interesting future work, and will add it to Appendix F.
>
> For Slot-TTA, we conducted preliminary experiments by optimizing the denoising loss of SlotDiffusion on input images at test time. We use the default hyper-parameters listed in the Slot-TTA paper. Below we show the segmentation results on CLEVRTex. Slot-TTA consistently improves the performance of SlotDiffusion across all metrics, demonstrating the ability of our method to benefit from recent advances in slot-based models.
>
> |CLEVRTex|FG-ARI &uarr;|mIoU &uarr;|mBO &uarr;|
> |-|-|-|-|
> |SlotDiffusion|68.39|57.56|61.94|
> |SlotDiffusion + Slot-TTA|**69.66**|**58.67**|**62.82**|
>
> [4] Prabhudesai, Mihir, et al. "Test-time adaptation with slot-centric models." ICML. 2023.

---

> > ### Comment · Reviewer_QakT · 2023-08-11
> >
> > Thanks for the detailed response + experiments.
> > I hope the authors include the new extensions (Slot-TTA and BO-SlotDiffusion) in their main paper.
> >
> > Further i would encourage the authors to add comparisions with CutLer in their supplementary.  This would encourage more work in this direction.
> >
> > I have increased my final rating due to the added experiments.

---

> > > ### Author Response · Authors · 2023-08-11
> > > **Thank you!**
> > >
> > > Thank you for updating your review! We are happy to hear that we were able to address all of your concerns. We will include the results in the final version of the paper as you suggested.

---

### Author Rebuttal · Authors · 2023-08-08

We would like to thank the reviewers for their helpful feedback and insightful comments.

We are glad that the reviewers find our paper “*well written*” (QakT), “*easy to follow*” (GFM9), and “*clear*” (jjQP), our diffusion-based slot decoder as “*sound*” (GFM9) and “*novel*” (jjQP). Also, our experiments are considered “*dense*” (QakT), “*rigorous*” (GFM9), and “*provides fruitful results that effectively validate the proposed method from various perspectives*” (JU9y).

From the view of object-centric learning, reviewer GFM9 remarks that our work “*fills a gap in the literature*” and “*is a valuable contribution*” to the community, and highlights our insight of treating slots as text features as “*novel and potentially quite powerful*”. Reviewer jjQP finds SlotDiffusion “*comparing favourably against*” other slot-based models “*in their respective settings*”. Finally, reviewer JU9y agrees that “*This direction of slot-based generation is promising*”.

We further thank the reviewers for their constructive feedback. We have uploaded a PDF file which includes figures and large tables to address the reviewers’ feedback. **Overall, we want to highlight two new experiments we conducted:**

**1. Combining SlotDiffusion with recent improvements in slot-based methods.** (suggested by reviewer QakT and reviewer jjQP)

We combine SlotDiffusion with two recent improvements on slot-based models, Slot-TTA [1] and bi-level optimization (dubbed BO-SlotDiffusion) [2], and report the results on CLEVRTex in the tables below.

|CLEVRTex|FG-ARI &uarr;|mIoU &uarr;|mBO &uarr;|
|-|-|-|-|
|SlotDiffusion|68.39|57.56|61.94|
|SlotDiffusion + Slot-TTA|**69.66**|**58.67**|**62.82**|

|CLEVRTex|FG-ARI &uarr;|mIoU &uarr;|mBO &uarr;|MSE &darr;|LPIPS &darr;|FID &darr;|
|-|-|-|-|-|-|-|
|BO-QSA|**80.47**|64.26|66.27|268.2|0.38|98.93|
|SlotDiffusion|68.39|57.56|61.94|237.5|0.13|32.07|
|BO-SlotDiffusion|78.50|64.35|**68.68**|240.3|0.13|**30.06**|

See our response to individual reviews for detailed analysis (Slot-TTA: **point 5** to reviewer QakT; bi-level optimization: **point 2** to reviewer jjQP). Overall, our method is able to benefit from recent advances in slot-based methods, and further boost its performance.

[1] Prabhudesai, Mihir, et al. "Test-time adaptation with slot-centric models." ICML. 2023.

[2] Jia, Baoxiong, et al. "Improving object-centric learning with query optimization." ICLR. 2023.


**2. Basic comparison between SlotDiffusion and concurrent work LSD [3]** (suggested by reviewer JU9y)

Since LSD has not released its code, we can only compare with it on CLEVRTex and MOVi-E where we both conduct experiments. We try our best to re-train and test SlotDiffusion under the same setting as LSD. We report the results in **Table 2 of the submitted PDF file**.

On CLEVRTex, we achieve better FG-ARI, and comparable results in mIoU, mBO, and FID. On MOVi-E, we outperform LSD in both segmentation and generation metrics. Please see our response to reviewer JU9y (**point 3**) for more details.

[3] Jiang, Jindong, et al. "Object-centric slot diffusion." arXiv preprint arXiv:2303.10834 (2023).

For other questions raised by the reviewers, please see our response (with text and other new experiments) to individual questions below each review. We will incorporate all our responses and additional results in the final version of the manuscript.

---

### Decision · Program_Chairs · 2023-09-21

**Decision:**

Accept (spotlight)

**Comment:**

All reviewers reached a consensus to accept the paper with review scores of 6/7/6/7.

The paper proposes a diffusion-based decoder for slot-centric models, which results in better generation, segmentation and video prediction. All reviewers acknowledged the novelty and importance of the work. They also found the paper solid and well-written.

The authors provided additional experimental results and clarifications in the rebuttal, which the reviewers found very helpful. They are suggested to include the new extensions (Slot-TTA and BO-SlotDiffusion) and the discussion about scaling the decoders from prior work in their main paper. They are also encouraged to add the comparisons with CutLer in their supplementary.